# Histone H3K23-specific acetylation by MORF is coupled to H3K14 acylation

Brianna J. Klein[1,12], Suk Min Jang[2,12], Catherine Lachance[2], Wenyi Mi[3], Jie Lyu [4,5], Shun Sakuraba [6], Krzysztof Krajewski[7], Wesley W. Wang[8], Simone Sidoli [9,11], Jiuyang Liu[1], Yi Zhang [1], Xiaolu Wang[3], Becka M. Warfield[1], Andrew J. Kueh[10], Anne K. Voss[10], Tim Thomas[10], Benjamin A. Garcia[9], Wenshe R. Liu [8], Brian D. Strahl [7], Hidetoshi Kono [6], Wei Li [4,5], Xiaobing Shi [3], Jacques Côté [2]* & Tatiana G. Kutateladze [1]*

Acetylation of histone H3K23 has emerged as an essential posttranslational modification associated with cancer and learning and memory impairment, yet our understanding of this epigenetic mark remains insufficient. Here, we identify the native MORF complex as a histone H3K23-specific acetyltransferase and elucidate its mechanism of action. The acetyltransferase function of the catalytic MORF subunit is positively regulated by the DPF domain of MORF (MORF$_{DPF}$). The crystal structure of MORF$_{DPF}$ in complex with crotonylated H3K14 peptide provides mechanistic insight into selectivity of this epigenetic reader and its ability to recognize both histone and DNA. ChIP data reveal the role of MORF$_{DPF}$ in MORF-dependent H3K23 acetylation of target genes. Mass spectrometry, biochemical and genomic analyses show co-existence of the H3K23ac and H3K14ac modifications in vitro and co-occupancy of the MORF complex, H3K23ac, and H3K14ac at specific loci in vivo. Our findings suggest a model in which interaction of MORF$_{DPF}$ with acylated H3K14 promotes acetylation of H3K23 by the native MORF complex to activate transcription.

[1] Department of Pharmacology, University of Colorado School of Medicine, Aurora, CO 80045, USA. [2] Laval University Cancer Research Center, CHU de Québec-UL Research Center-Oncology Division, Quebec City, QC G1R 3S3, Canada. [3] Center for Epigenetics, Van Andel Research Institute, Grand Rapids, MI 49503, USA. [4] Department of Biological Chemistry, University of California, Irvine, Irvine, CA 92697, USA. [5] Dan L. Duncan Cancer Center, Department of Molecular and Cellular Biology, Baylor College of Medicine, Houston, TX 77030, USA. [6] Molecular Modeling and Simulation Group, National Institute for Quantum Life Science, National Institutes for Quantum and Radiological Science and Technology, Kizugawa, Kyoto 619 0215, Japan. [7] Department of Biochemistry & Biophysics, The University of North Carolina School of Medicine, Chapel Hill, NC 27599, USA. [8] Department of Chemistry, Texas A&M University, College Station, TX 77843, USA. [9] Epigenetics Program, Department of Biochemistry and Biophysics, Perelman School of Medicine, University of Pennsylvania, Philadelphia, PA 19104, USA. [10] The Walter and Eliza Hall Institute of Medical Research, Parkville, VIC 3050, Australia. [11] Present address: Department of Biochemistry, Albert Einstein College of Medicine, New York, NY 10461, USA. [12] These authors contributed equally: Brianna J. Klein, Suk Min Jang. *email: jacques.cote@crhdq.ulaval.ca; tatiana.kutateladze@cuanschutz.edu

Acetylation of lysine residues in histone proteins is a canonical mechanism by which eukaryotic cells regulate chromatin structure and function. Lysine acetylation is widely associated with an open and transcriptionally active state of chromatin. As a posttranslational modification (PTM), acetyllysine weakens electrostatic interactions between histones and DNA, resulting in less compact chromatin and more accessible DNA, and provides binding sites for proteins and complexes that promote gene expression[1,2]. The level and sites of lysine acetylation are tightly regulated through the opposing enzymatic activities of histone acetyltransferases (HAT) that generate this mark and histone deacetylases that remove it[3]. Five acetyltransferases, including monocytic leukemic zinc-finger related factor (MORF), MOZ, HBO1, hMOF and Tip60 (also known as KAT6B, KAT6A, KAT7, KAT8 and KAT5, respectively) are characterized by the catalytic MYST domain and together comprise one of the three major families of human HATs, the MYST family[4].

The MORF protein is a core catalytic subunit of the native complex which bears the same name (MORF) and has been implicated in acetylation of histone H3 and H4 [5,6]. Specifically, MOZ/MORF complexes were reported to acetylate H3K14 and H4K5/8/12/15 in vitro and H3K9 in vivo[7–12]. The acetyltransferase activity of the MORF protein is essential in neurogenesis and skeletogenesis, whereas chromosomal translocations and mutations in MORF are linked to aggressive forms of cancer and developmental diseases[5,6,13]. Truncated isoforms of MORF are found in Say−Barber−Biesecker−Young−Simpson and Genitopatellar syndromes, which are associated with intellectual disabilities[14,15]. Inhibition of the MORF HAT activity has been shown to induce senescence and arrest tumor growth[16]. Although MORF has many overlapped functions with its paralog MOZ, these proteins are also involved in distinct developmental programs and reciprocally regulated in macrophage activation pathways[17].

The histone sites H3K9 and H3K14 have long been reported as substrates of MOZ and to some extent of MORF[6]. More recent studies in glioma cells show that MOZ is capable of generating H3K23ac, which recruits TRIM24 and subsequently activates a signaling pathway that facilitates glioblastoma, ER-driven breast cancer, and AR-driven prostate cancer[18–21]. The *Drosophila* homolog, Enok, also acetylates H3K23 and regulates genes responsible for germline cell formation and abdominal segmentation in fly embryos[22]. Accumulated clinical and cell experimental data point to a direct link between the MORF complex deregulation and developmental disorders and cancer, yet the precise biological function of this complex remains unclear.

In addition to the catalytic MORF subunit, the MORF complex is thought to contain three other core subunits, i.e., BRPF1/2/3, ING5 and MEAF6[5,23] (Fig. 1a, b). The MORF subunit consists of the NEMM (N-terminal part of Enok, MOZ or MORF) domain, the double PHD finger (DPF), the MYST domain, and the ED (glutamate/aspartate-rich) and SM (serine/methionine-rich) regions (Fig. 1a). Whereas the precise function of the NEMM domain is not well understood, some sequence similarity to histones H1 and H5 suggests a regulatory role[24]. The DPF module of MORF has been shown to bind histone H3 tail acylated at lysine 14 (H3K14acyl)[25–27]. The catalytic MYST domain acetylates lysine residues in histone and non-histone substrates and interacts with BRPF1/2/3[5,28]. The ED and SM regions have been proposed to have a role in transcriptional activity of the complex[5].

In this study, we show that the native MORF complex, containing BRPF1, ING4/5 and MEAF6, acetylates H3K23 and that this catalytic activity is coupled to binding of the DPF domain of MORF to another acylated, particularly crotonylated, modification—H3K14cr. We further report on substantial co-existence of the two acetylation sites, H3K23 and H3K14, in vitro and in vivo and show that both modifications colocalize with the MORF complex at target genes.

## Results

**The native MORF complex acetylates histone H3K23.** Mammalian MORF complexes have been identified through affinity purification using associated proteins as bait, transient cotransfections, or through baculovirus-mediated expression[10,24,29,30]. To characterize endogenous native forms of the complexes, we engineered K562 cell lines[31] that express epitope tagged (3xFLAG-2xStrep) MORF and paralogous MOZ from the *AAVS1* safe harbor. Tandem affinity purification of both HAT proteins confirmed the tetrameric nature of the MORF and MOZ complexes but also identified BRPF1 as the sole scaffold subunit of these complexes (Fig. 1c and Supplementary Data 1). Formation of the native MORF-BRPF1-ING4/5-MEAF6 complex was confirmed by Western blot analysis (Fig. 1d). The acetyltransferase activity of the MORF complex was then assessed on unmodified and acetylated histone H3 peptides (residues 1–29 of H3). The MORF complex acetylated unmodified H3 peptide, but its catalytic activity was considerably, by ~12-fold, decreased on the H3 peptide that was pre-acetylated at Lys23 (H3K23ac) (Fig. 1e, blue bars). In contrast, HAT activity of the complex on the peptide pre-acetylated at Lys14 (H3K14ac) was reduced only ~1.2-fold. These results suggest that H3K23 is a preferential acetylation site of the complex.

To increase purification yield, we generated the MORF complex consisting of BRPF1, ING5, MEAF6 and FLAG-MORF$_N$ (residues 1–716 of MORF) via co-transfection of the subunits in 293T cells and measured enzymatic activity of the complex (Fig. 1f and Supplementary Fig. 1a). Much like the native MORF complex, the MORF complex with cotransfected subunits was capable of acetylating unmodified H3 peptide. The HAT activity of this complex was only slightly, by ~1.2-fold, decreased on H3K14ac peptide, implying that H3K14 is not its primary substrate. Again, a substantial ~7-fold decrease in the catalytic activity of the complex was observed on H3K23ac peptide, supporting the idea that H3K23 is the major target for acetylation (Fig. 1g).

The catalytic MYST domain of the MORF subunit is preceded by the DPF module (MORF$_{DPF}$) (Fig. 1a) that was previously shown to recognize histone H3K14ac[26,27]. To determine whether MORF$_{DPF}$ influences the enzymatic function of the MORF complex, we deleted the DPF domain in the MORF subunit (MORF ΔDPF) expressed from the AAVS1 locus, purified the corresponding native complex, and assayed its acetyltransferase activity (Fig. 1e, black bars and Supplementary Fig. 1b, c and 2). The HAT activity of the MORF ΔDPF complex was decreased ~2-fold on the H3 and H3K23ac peptides and on purified free histones, and ~4-fold on the H3K14ac peptide, indicating that DPF affects the catalytic function of the MORF complex.

**MORF$_{DPF}$ recognizes acylated H3K14 and prefers crotonylation.** The DPF domains of MOZ and DPF2 have recently been shown to select for the crotonylated modification H3K14cr over other acylated H3K14 species in vitro and in vivo[32,33]. It was also found that crotonylation is associated with highly transcriptionally active chromatin[32,34]. We examined whether the selectivity toward H3K14cr is conserved in MORF$_{DPF}$ using a combination of histone peptide pulldown assay, NMR and fluorescence spectroscopy. The pulldown assay confirmed that MORF$_{DPF}$ recognizes the N-terminal tail of histone H3 (residues 1–22 of H3), and H3K14 crotonylation indeed augmented this

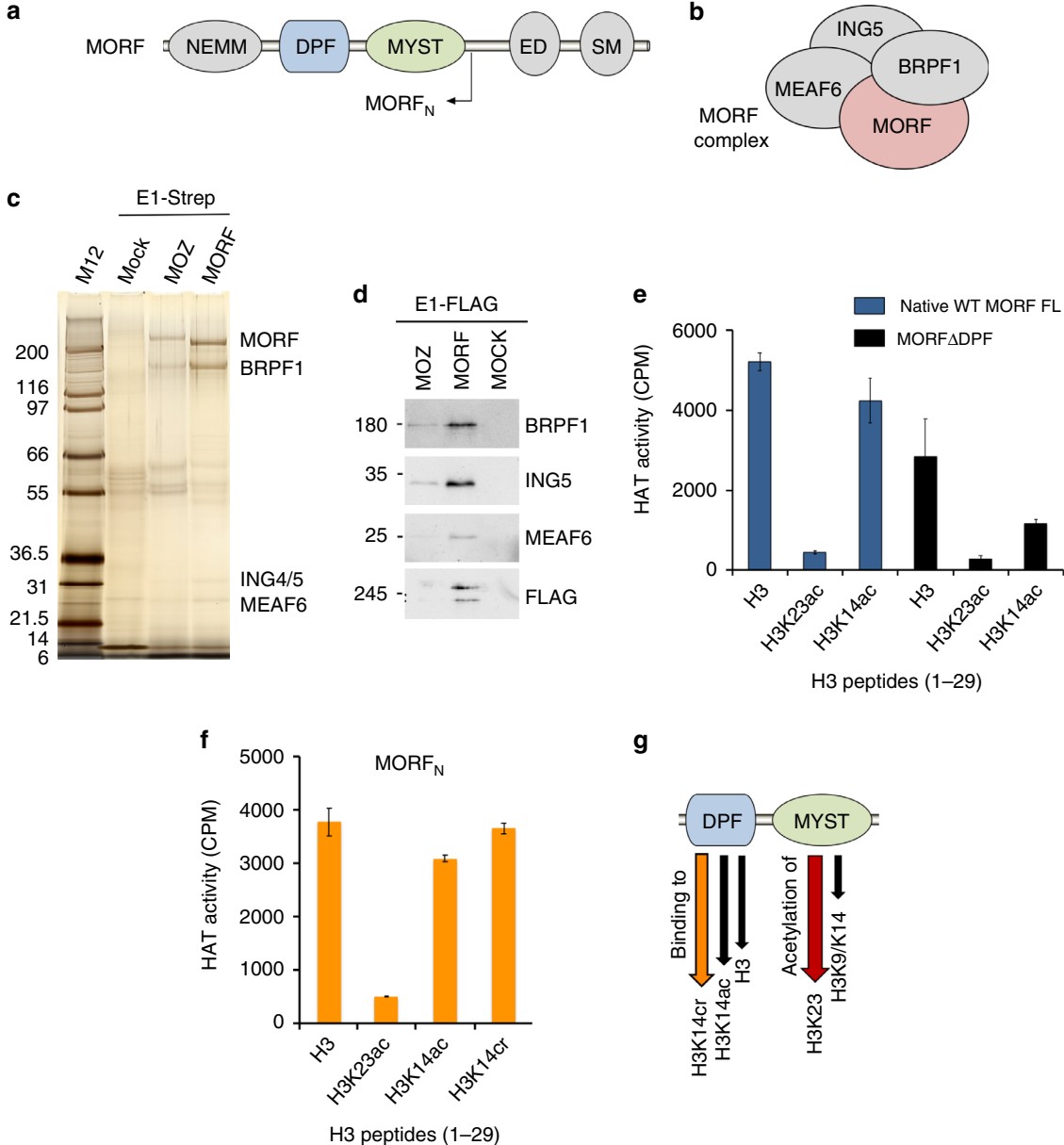

**Fig. 1** The native MORF complex is a H3K23-specific HAT. **a** A diagram of MORF with the MORF$_N$ region indicated. **b** The MORF complex subunit composition. **c** Purified native MOZ and MORF complexes from K562 cells shown by silver staining. **d** Confirmation of the subunits of the native purified MORF complex by Western blots using specific antibodies. Source data are provided as a Source Data file. **e** HAT activity of native MORF complex from K562 cells on the indicated H3 peptides. Liquid HAT assays, in which reactions were spotted on P81 filters and counted by scintillation as counts per minute (CPM). **f** HAT activity of the MORF$_N$ complex overexpressed in 293T cells on indicated H3 peptides. **g** Cartoon showing interrelated activities of the adjacent DPF and MYST domains. The MYST domain acetylates primarily H3K23 and has some HAT activity on H3K14, whereas MORF$_{DPF}$ binds to unmodified H3 or acylated H3K14, preferring H3K14cr (see below). Error bars indicate the range from duplicate samples, $n = 2$ independent experiments

interaction (Fig. 2a). Dissociation constants ($K_d$s) measured by tryptophan fluorescence revealed that MORF$_{DPF}$ binds ~3-fold tighter to H3K14cr peptide ($K_d = 0.7\,\mu M$) than to H3K14ac peptide ($K_d = 1.8\,\mu M$) (Fig. 2b, c and Supplementary Fig. 3). The interaction of MORF$_{DPF}$ with H3K14cr peptide was further corroborated by substantial chemical shift perturbations observed in NMR $^1$H,$^{15}$N heteronuclear single quantum coherence (HSQC) titration experiments in which unlabeled H3K14cr (residues 1–19 of H3) peptide was titrated in $^{15}$N-labeled MORF$_{DPF}$ (Fig. 2d).

To gain insight into the molecular mechanism underlying recognition of crotonyllysine, we co-crystallized MORF$_{DPF}$ with H3K14cr peptide (residues 1–19 of H3) and refined the structure

of the complex to 2.1 Å resolution (Fig. 2e and Supplementary Table 1). The overall structure of MORF$_{DPF}$ bound to H3K14cr peptide is very similar to the previously determined structure of this domain in complex with H3K14bu peptide[14]. A comparable set of intermolecular hydrogen bonds and electrostatic contacts stabilize both complexes. Residues R2-T3 of the H3K14cr peptide are in an extended conformation, whereas K4-T11 residues adopt an α-helical conformation. The structure of the MORF$_{DPF}$-H3K14cr complex superimposes with the structures of the MOZ$_{DPF}$-H3K14cr[32] and MORF$_{DPF}$-H3K14bu[14] complexes with RMSDs of 0.4 Å and 0.5 Å, respectively, indicating that the acyllysine-binding mechanism is in general conserved.

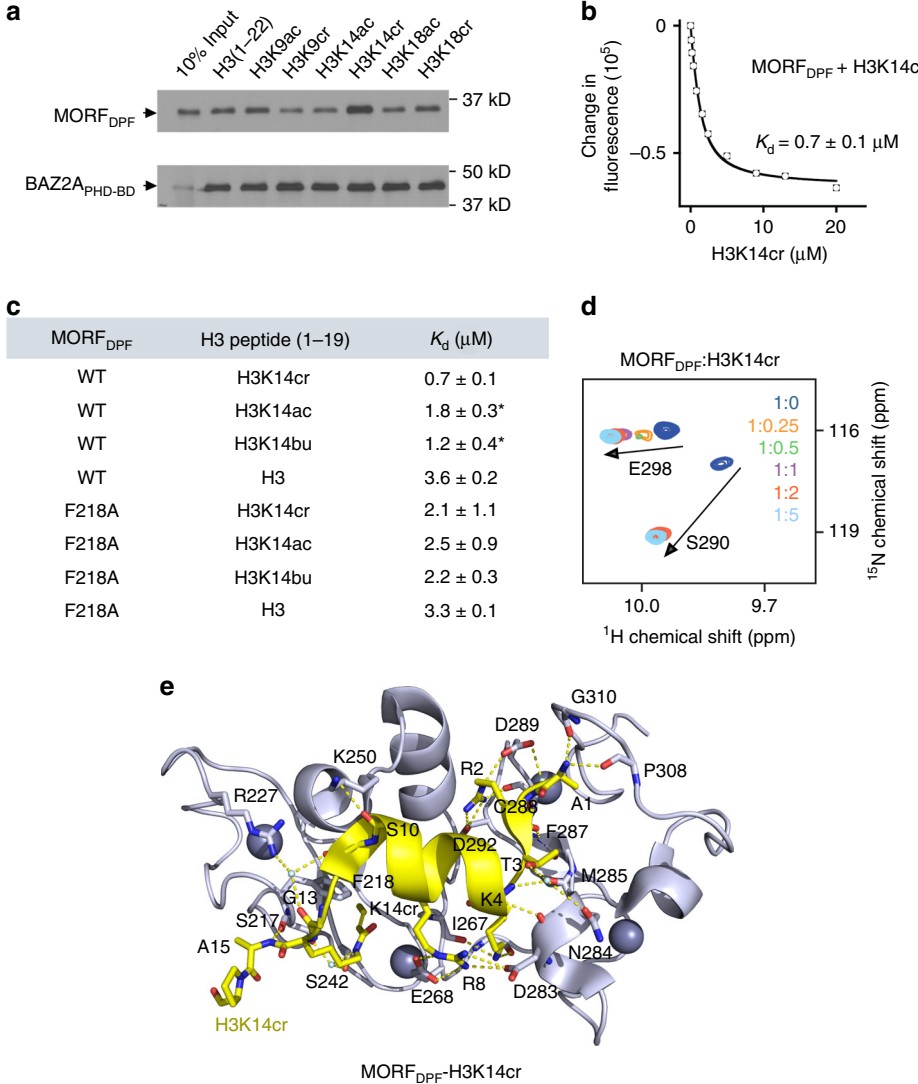

**Fig. 2** MORF_DPF recognizes H3K14cr. **a** Western blot analysis of peptide pulldown experiments of GST-MORF_DPF with the indicated histone H3 peptides. Peptide pulldowns of the BAZ2A PHD-Bromo domains are shown for comparison. **b** Representative binding curve used to determine $K_d$ by tryptophan fluorescence. **c** Binding affinities of MORF_DPF to the indicated peptides as measured by tryptophan fluorescence. (*) from[26,27] $K_d$ values were averaged over at least three separate experiments (two for F218A/H3), and error was calculated as standard deviation (SD) between the runs. **d** Superimposed $^1$H,$^{15}$N HSQC spectra of $^{15}$N-labeled MORF_DPF, collected as H3K14cr peptide was added stepwise. Spectra are color coded according to the protein:peptide molar ratio (inset). **e** The ribbon diagram of the MORF_DPF-H3K14cr complex structure. H3K14cr peptide is yellow. The histone peptide residues and the residues of MORF_DPF involved in the interaction are labeled. Dashed lines indicate hydrogen bonds. Source data are provided as a Source Data file

**A ping-pong-like acyllysine-binding mechanism**. To characterize the mechanistic basis of the MORF_DPF selectivity for H3K14cr, we compared the structures of MORF_DPF in complex with H3K14cr and H3K14bu[27]. The butyryl and crotonyl moieties are similar in size but unlike the saturated butyryl group, the crotonyl group contains a C=C double bond. In the structures, K14cr and K14bu occupy the same hydrophobic channel, formed by S217, F218, L235, S242, S243, G244, W264, C266 and I267 residues of MORF_DPF, and are stabilized through a water-mediated hydrogen bond involving the side-chain amide nitrogen atom of acyllysine and the hydroxyl groups of S217 and S242 (Fig. 3a and Supplementary Fig. 4). Both acyl chains are positioned in close proximity to the aromatic ring of F218. We postulated that if F218 and the C=C double bond are involved in a π−π interaction, it would strengthen the association of MORF_DPF with H3K14cr but not with H3K14bu. Indeed, replacement of F218 with alanine eliminated selectivity of MORF_DPF toward

H3K14cr. Comparable $K_d$ values were measured for the interaction of the MORF_DPF F218A mutant with H3K14cr or H3K14bu (Fig. 2c and Supplementary Fig. 3), implying that F218 is the main driving force of selectivity for the unsaturated acyl group.

Although the distances between the sp$^2$ carbon atoms of the crotonyl and F218 π systems in the crystal structure of the MORF_DPF-H3K14cr complex are short (3.8 Å and 4.2 Å, Fig. 3a), the mutual orientation of these systems is not ideal to insure overlap of the p orbitals and thus substantially contribute to the enthalpy (Fig. 3a). We carried out the molecular dynamics (MD) simulations to determine the extent of the H3K14cr-binding site flexibility. Analysis of MD trajectories revealed that H3K14cr is stabilized by a pair of switching hydrogen bonds that allow crotonyllysine to shuffle between two conformations in a ping-pong manner (Fig. 3b–d and Supplementary Fig. 5). The side-chain amide nitrogen atom of crotonyllysine was in close contact with the hydroxyl oxygen atom of S217 but was away from I267

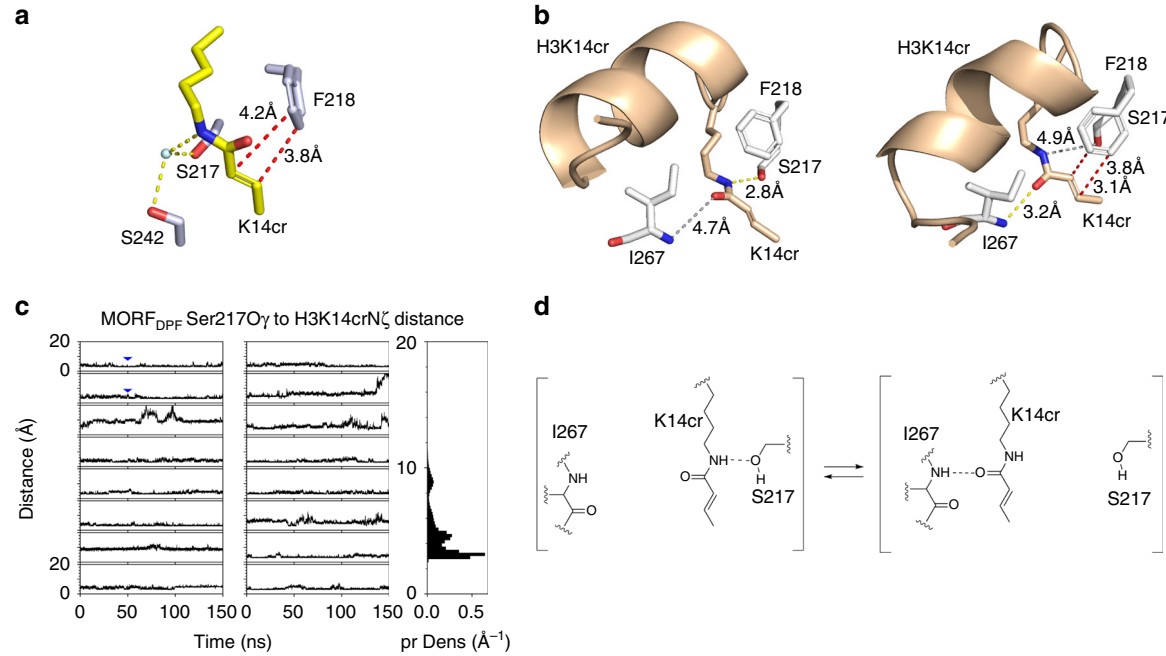

**Fig. 3** The ping-pong-like acyllysine-binding mechanism. **a** A zoom-in view of the H3K14cr-binding pocket of MORF$_{DPF}$. Yellow dashed lines represent water (cyan)-mediated hydrogen bonds. Red dashed lines represent short distances indicative of the $\pi-\pi$ interaction. **b** A close view of the H3K14cr-binding site of the MD generated complex. Yellow and red dashed lines represent hydrogen bonds and short distances indicative of the $\pi-\pi$ interaction, respectively. Gray lines show long distances between the indicated residues. **c** (left two columns) Time-course of the distance between Ser217O$\gamma$ of MORF$_{DPF}$ and K14crN$\zeta$ of H3. Blue triangles correspond to the timing of snapshots shown in **b**. All plots share the same ordinate (between 0 and 20 Å). (Right column) Distribution of the distance between two atoms. Abscissa and ordinate represent the probability density (pr Dens) and the distance between Ser217O$\gamma$ of MORF$_{DPF}$ and K14crN$\zeta$ of H3. **d** A schematic showing shuffling of K14cr between two conformations in the MORF$_{DPF}$-H3K14cr complex

in one conformation (Fig. 3b, left). In another conformation, the side-chain carbonyl oxygen of crotonyllysine was in close contact with the backbone nitrogen atom of I267 but shifted away from S217 (Fig. 3b, right). In addition to slowing an off rate due to ping-pong shuffling, the crotonyllysine-I267 contact aligned the crotonyl and F218 $\pi$ systems for a more efficient $\pi-\pi$ interaction. MD simulations of the MORF$_{DPF}$-H3K14bu complex showed that butyryllysine is also capable of shuttling between the conformations. Together, these results suggest that the ping-pong-like acyllysine-binding mechanism might be a characteristic feature of the MORF$_{DPF}$-H3K14acyl complex formation that can explain preference of MORF$_{DPF}$ for the acylated H3K14 species.

**H3K23 acetylation depends on histone binding by MORF$_{DPF}$.** To determine the effect of histone binding by MORF$_{DPF}$ on H3K23 acetylation, we produced the MORF complexes harboring a previously identified loss-of-function mutation in MORF$_{DPF}$ (ref. [26]) in the AAVS1-integrated full-length MORF or transfected MORF$_N$ subunit and tested these complexes in HAT assays using free histones, histone H3 peptides, and recombinant NCPs as substrates. NMR titration experiments confirmed that substitution of F287 with alanine decreases binding of MORF$_{DPF}$ to H3K14cr, H3K14ac or H3 peptides ~$10^3$-fold compared to binding of the WT protein to the same peptides (Fig. 4a, b and Supplementary Fig. 6a). The catalytic activity of the full-length MORF F287A complex on free histones was reduced two-fold compared to the HAT activity of the native WT MORF complex, indicating that functional MORF$_{DPF}$ is essential (Fig. 4c and Supplementary Fig. 1d). A similar two-fold reduction in the HAT activity of the MORF$_N$ F287A complex was observed on H3K14cr peptide (Fig. 4d, blue bars). Both WT MORF$_N$ and MORF$_N$

F287A complexes had substantially reduced residual enzymatic activity on H3K23ac peptide, confirming that MORF is a H3K23-specific acetyltransferase (Fig. 4d, black bars). The decrease in HAT activity of the MORF$_N$ F287A complex on H3 peptide further pointed to the role of histone binding by MORF$_{DPF}$ in the catalytic function of the MORF complex (Fig. 4e and Supplementary Fig. 7). A less pronounced reduction in the HAT activity of the MORF$_N$ F287A complex on recombinant NCP and H3K14ac-NCP (Fig. 4d, green and orange bars) indicated that other readers present in the complex, including the PZP domain of BRPF1 that binds to H3 tail and DNA[35], tether the complex to NCPs, counterbalancing the loss of the interaction with MORF$_{DPF}$. We also note that in comparison with H3K14ac peptide, H3K14cr peptide was a better substrate for the MORF$_N$ complex (Fig. 1f), suggesting that recognition of the crotonyllysine modification by MORF$_{DPF}$ can promote acetylation by the MORF complex.

**MORF$_{DPF}$ has DNA-binding function.** Electrostatic surface potential of MORF$_{DPF}$ reveals positively charged surface regions, particularly encompassing residues R306/K309 and R276/R293, which could potentially interact with negatively charged DNA (Fig. 5a). To determine whether MORF$_{DPF}$ is capable of binding to DNA, we examined its association with 147 bp 601 DNA in electrophoretic mobility shift assay (EMSA). Increasing amounts of MORF$_{DPF}$ were incubated with 601 DNA, and the reaction mixtures were resolved on a native polyacrylamide gel (Fig. 5b). A gradual increase of the MORF$_{DPF}$ concentration led to the shift of DNA band, indicating that wild-type MORF$_{DPF}$ forms complex with DNA (Fig. 5b). To map the DNA-binding site, we produced K309E/R306E and R276E mutants of MORF$_{DPF}$ and tested them in

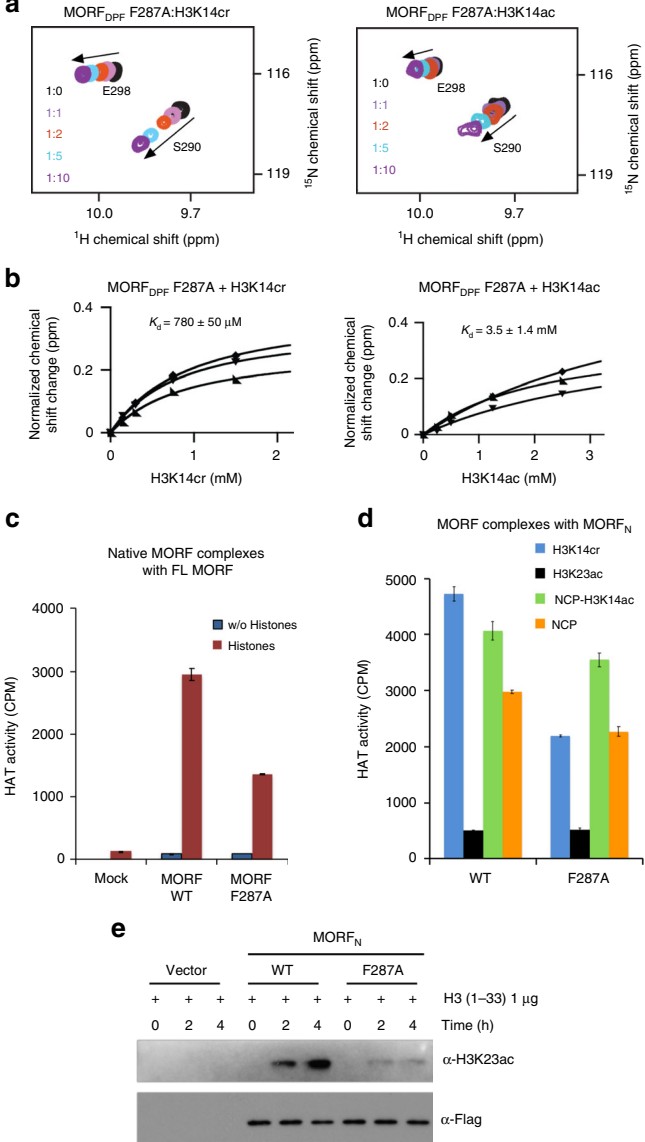

**Fig. 4** H3K23 acetylation depends on functional MORF_DPF. **a** Superimposed $^1$H,$^{15}$N HSQC spectra of MORF_DPF F287A mutant collected as indicated peptides were added stepwise. Spectra are color coded according to the protein:peptide molar ratio (inset). **b** Representative binding curves used to determine $K_d$s for the interactions of MORF_DPF F287A with indicated peptides by NMR. Error was calculated as SD. **c** HAT activity of native MORF wild-type and F287A mutant complexes from K562 cells on native free core histones. Liquid HAT assays, in which reactions were spotted on P81 filters and counted by scintillation as counts per minute (CPM). **d** HAT activities of wild-type and mutated MORF_N complexes purified from 293T cells using indicated H3 peptides (residues 1–29) or recombinant unmodified nucleosomes (NCP) or nucleosomes containing H3K14ac (H3K14ac-NCP) as substrates. Error bars in **c** and **d** indicate the range from duplicate samples, $n = 2$ independent experiments. **e** Western blot analysis of HAT assays with purified overexpressed wild-type and mutant MORF_N using H3 peptide (residues 1–33) as substrate

EMSA (Fig. 5c, d and Supplementary Fig. 8). While the MORF_DPF R276E mutant retained the DNA-binding ability, binding of the K309E/R306E mutant to 601 DNA was decreased, suggesting that the region of MORF_DPF containing R306 and K309 is involved in the interaction with DNA (Supplementary Fig. 8).

To assess the contribution of histone-binding and DNA-binding activities of MORF_DPF in chromatin targeting, we reconstituted H3K14cr-containing nucleosome core particles (H3K14cr-NCP) and tested binding of wild-type and mutated MORF_DPF to these NCPs in EMSA. Incubation of increasing amounts of wild-type GST-MORF_DPF with H3K14cr-NCP resulted in disappearance of the H3K14cr-NCP band, suggesting formation of the GST-MORF_DPF:H3K14cr-NCP complex (Fig. 5c–e). The GST-MORF_DPF F287A mutant with impaired histone binding activity caused a moderate decrease in binding to H3K14cr-NCP (Fig. 5c, right), whereas the MORF_DPF R306E/K309E mutant defective in DNA binding was unable to associate with H3K14cr-NCP (Fig. 5d, right). These data indicate that while both interactions of MORF_DPF with DNA and histone H3 tail are essential for binding to the nucleosome, interaction with DNA contributes greater. Of note, another loss of histone binding function mutant of MORF_DPF, D289A, associated with H3K14cr-NCP tighter than wild-type MORF_DPF (Fig. 5e). The side chain of D289 is critical in recognition of H3: it restrains positively charged Arg2 and Ala1 (NH$_3^+$) of H3; however, D289 is located in close proximity to the DNA-binding R306/K309 patch (Fig. 5a). Thus, elimination of the negative charge near the DNA-binding site via substituting D289 with alanine most likely accounts for the enhanced binding of the D289A mutant to DNA within H3K14cr-NCP. Conversely, replacing two positive charges with negative charges near the H3 binding pocket (mutating R306 and K309 to Glu) resulted in a ~3-fold increase in binding to H3K14cr peptide by the F287A/R306E/K309E mutant compared to the F287A mutant (Fig. 4a, b, left panels, and Supplementary Fig. 6b). Collectively, these results demonstrate that DPF of MORF is characterized by a unique setting of the closely positioned H3K14acyl-binding and DNA-binding sites that provide a fine-tuned balance of electrostatic contacts with chromatin (Fig. 5a, f and Supplementary Fig. 6c).

**Functional MORF_DPF is required for H3K23 acetylation in vivo.** To examine the role of MORF_DPF in genomic occupancy of MORF and MORF-dependent H3K23 acetylation in cells, we performed chromatin immunoprecipitation experiments (ChIP) (Fig. 6a–c and Supplementary Data 2). 293T cells expressing FLAG-tagged wild-type or mutated MORF_N were used to measure MORF, H3K23ac and H3K14ac levels on promoters of a set of target genes. As shown in Fig. 6a, we observed a decreased association of FLAG-MORF F287A mutant defective in histone H3 binding with promoters compared to WT FLAG-MORF. Furthermore, ChIP H3K23ac and H3K14ac data revealed that expression of WT FLAG-MORF but not F287A mutant increases H3K23ac level on these promoters (Fig. 6b), whereas H3K14ac level was unaffected by the mutation in MORF_DPF (Fig. 6c). Together, these results indicate that functional MORF_DPF is required for proper localization of MORF in vivo and acetylation of H3K23 at target genes.

**MORF complex colocalizes with H3K23ac and H3K14ac in vivo.** To confirm that the MORF complex occupies genomic regions enriched in H3K23ac and H3K14ac, we analyzed ChIP-sequencing (ChIP-seq) datasets generated in human RKO and HMEL cell lines[29,36]. The MORF complex core subunits BRPF1 and ING5 colocalized with H3K23ac and H3K14ac near promoters of target genes, suggesting that the MORF complex binds to these loci (Fig. 6d). Co-occupancy of the MORF subunits, H3K23ac, and H3K14ac at the same genomic sites supported our model in which binding of MORF_DPF to H3K14acyl promotes H3K23 acetylation by the MORF complex to activate transcription.

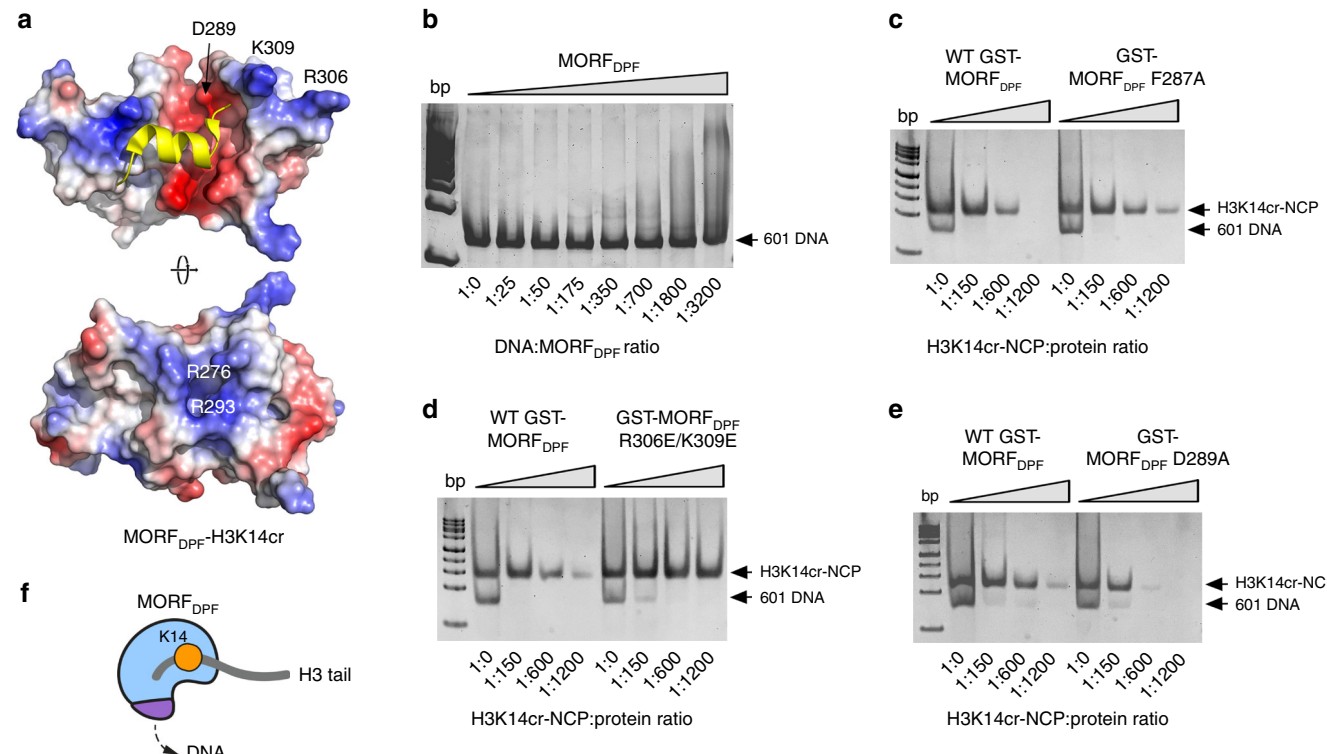

**Fig. 5** MORF$_{DPF}$ has DNA-binding activity. **a** Electrostatic surface potential of the MORF$_{DPF}$ in complex with H3K14cr is shown. Blue and red colors indicate positive and negative charges, respectively. **b** EMSA of 147-bp 601 DNA incubated with increasing amounts of WT MORF$_{DPF}$. DNA to protein molar ratio is shown below the gel image. **c–e** EMSA of H3K14cr-NCP incubated with increasing amounts of WT or mutant GST-MORF$_{DPF}$. H3K14cr-NCP to protein molar ratio is shown below each gel image. **f** A schematic showing the dual interaction of MORF$_{DPF}$ with histone tail and DNA

The extent of colocalization of H3K23ac and H3K14ac in vivo was assessed through analyzing ChIP-seq datasets generated by the ENCODE consortium in human IMR90 and hESC cell lines and the dataset generated in human HMEL cell line[36]. We observed enrichment of H3K23ac and H3K14ac near the transcription start sites (TSS) of genes in all these cell lines (Fig. 7a). Notably, the H3K23ac and H3K14ac signal intensities concomitantly increased with the increase in gene expression levels (Fig. 7b). Genome-wide localization analysis revealed a substantial overlap of H3K14ac-enriched and H3K23ac-enriched sites (Fig. 7c). For instance, in IMR90 cells, ~87% (27,669 out of 31,876) of the H3K23ac-enriched sites and ~56% (27,669 out of 49,063) of the H3K14ac-enriched sites overlapped.

**H3K14ac promotes H3K23 acetylation at specific genes**. To test if local H3K14 acetylation influences the activity of the MORF complex at specific genes in vivo, we carried out ChIP experiments in cells depleted of HBO1 (KAT7), the main enzyme responsible for H3K14 acetylation[37]. Hbo1$^{-/-}$ mouse embryonic fibroblasts[37] showed a near complete loss of H3K14ac and a concomitant significant loss of H3K23ac, but only at specific genes (Fig. 7d, e). In agreement, knockdown experiments using siRNA against HBO1 in human U2OS cells also demonstrated concomitant decreases of H3K14ac and H3K23ac at specific genes (Fig. 7f, g). These results indicate that H3K14ac deposition by HBO1 in vivo increases the local acetylation of H3K23, presumably through MORF (or MOZ), which acts as a reader and writer of these marks, respectively.

**H3K23ac and H3K14ac co-exist on the same histone tail**. While the individual H3K14 and H3K23 acylation marks have been identified[38,39], their co-existence remains uncharacterized. We used

middle-down mass spectrometry (MS) to analyze intact N-terminal tails from histone H3[40]. This approach allows for the identification of modified histone peptides of size of 50 residues and quantification of PTM co-frequencies. We calculated both the overall relative abundance of H3K23ac and H3K14ac by summing the relative abundance of all the quantified histone tails containing each of the two marks (Fig. 8a). These two marks were the two most abundant acetylations on histone H3—21.6% and 11.0% for H3K23ac and H3K14ac, respectively (Supplementary Data 3 and 4). We then quantified the co-frequency of the two PTMs, i.e., we summed the relative abundance of all the histone tails containing both marks (Fig. 8b). The H3K23ac and H3K14ac modifications had an overall co-existence frequency of 1.0%, which is the third most abundant co-existent combination of acetylation marks we detected on histone H3 (Supplementary Data 4).

## Discussion

Over the past decade remarkable progress has been made in the identification and characterization of acylation modifications in histone H3. The canonical H3K9, H3K14, H3K18 and H3K27 acylation sites have extensively been studied along with the enzymes that add and remove these marks. Much less is known regarding H3K23 acetylation, which has only recently been brought into the spotlight due to its role in oncogenesis and in learning and memory[18–20,41]. In this study, we characterized the native MORF complex, which was previously linked to multiple cancers and genetic disorders with intellectual disability, as a specific H3K23 acetyltransferase. Our findings are in agreement with results showing that BRPF1 KO or KD primarily affects H3K23ac levels in vivo[30,42]. Since the majority of BRPF1 is associated with MORF or MOZ in vivo[30], our data indicate that these complexes are the main source of H3K23ac in eukaryotes.

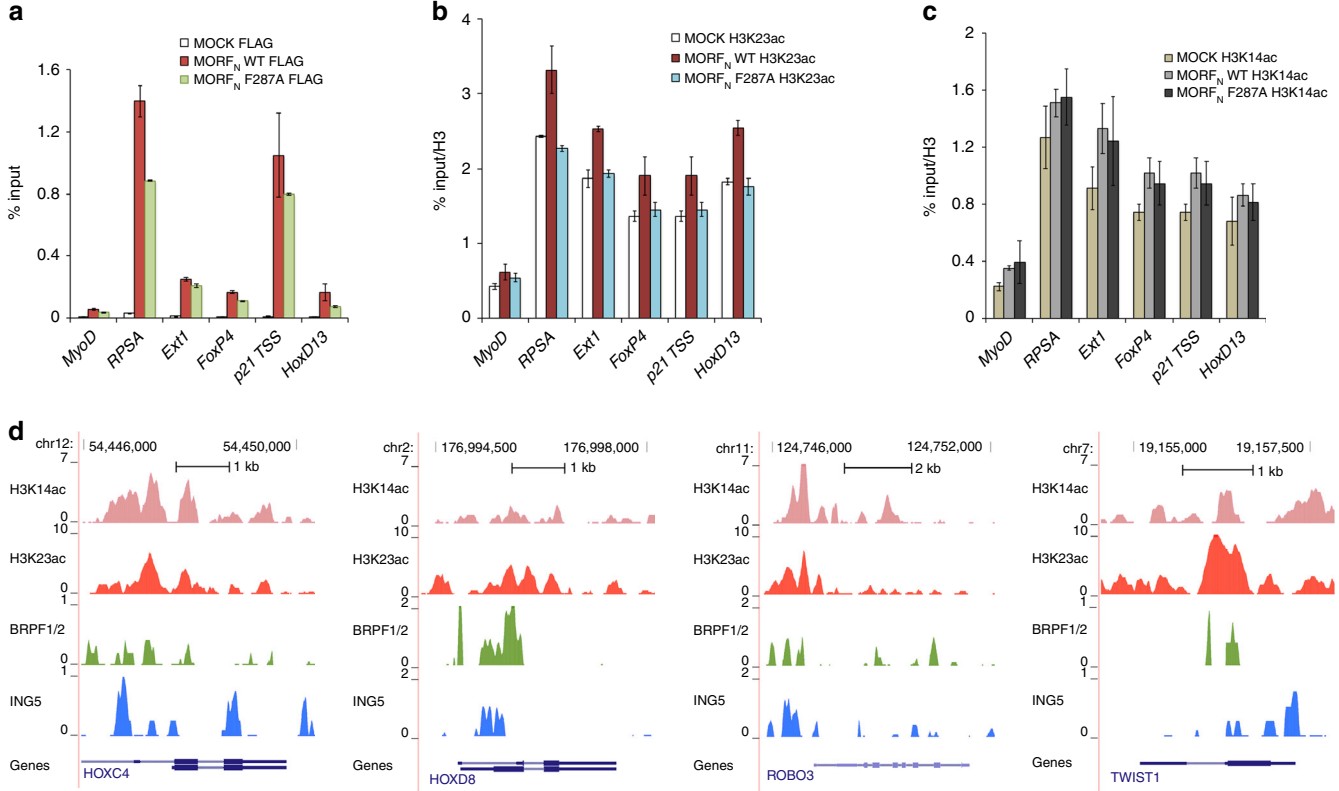

**Fig. 6** MORF$_{DPF}$ role in MORF localization in vivo and deposition of H3K23ac. **a** ChIP analysis of wild-type FLAG-MORF$_N$ and mutant FLAG-MORF$_N$ F287A in transfected 293T cells. **b**, **c** ChIP analysis of H3K23ac (**b**) and H3K14ac (**c**) in the same transfected cells as in **a**. Acetylation levels were corrected for nucleosomes occupancy (total H3 signal). All ChIP values are based on two independent experiments and error bars indicate the range between the samples. **d** ChIP-seq profiles of the MORF complex subunits (BRPF1/2 detected with the Brd1 Ab and ING5) in human RKO cells and of histone modifications (H3K14ac and H3K27ac) in HMEL cells are shown. The bottom tracks correspond to the representative RefSeq genes

We further show a high correlation between H3K23ac and H3K14ac in vitro and in vivo. Our data suggest that binding of MORF$_{DPF}$ to acylated H3K14 promotes acetylation of H3K23 by the native MORF complex. Because MORF$_{DPF}$ and the catalytic MORF$_{MYST}$ domain are linked, these domains could associate with the same H3 tail *in cis* or with two H3 tails *in trans*, whereas binding of MORF$_{DPF}$ to DNA could stabilize the complex on chromatin/nucleosomes (Fig. 8c). A model of MORF$_{DPF-MYST}$ bound to a single H3K14cr tail generated using the simulated annealing method and the crystal structures of MORF$_{DPF}$-H3K14cr and MOZ$_{MYST}$[9] reveals that the *cis* arrangement is ideal for the bivalent interaction (Supplementary Fig. 9). It will be interesting in future studies to explore this mechanism in detail and determine whether it is conserved in the MOZ complex. It will also be important to investigate the relationship between the MORF and BRPF1 subunits of the MORF complex, as the PZP domain of BRPF1 also binds to H3 tail and DNA[35].

## Methods

**Affinity purifications of MORF complexes.** Isogenic K562 cell lines (ATCC CCL-243) expressing 3xFLAG2xStrep-tagged MORF (wild type, delta DPF (Δ211–322) or F287A) and MOZ were generated by integration at the AAVS1 safe harbor locus after DSB induction and recombination targeted by co-transfection with a ZFN expression plasmid, as previously described[31]. 2e5 cells were transfected with 400 ng of ZFN expression vector and 4 μg of donor constructs. Selection and cloning were performed in RPMI medium supplemented with 0.5 μg/ml puromycin starting 2−3 days post transfection. Clones were obtained by limiting dilution and expanded before harvest for Western blot analysis.

For purification of native complexes, after large-scale expansion of K562 clones, tandem affinity purifications (TAP) of MORF and MOZ complexes were performed on nuclear extracts as previously described[43]. Briefly, nuclear extracts were prepared following standard procedures and precleared with CL6B Sepharose

beads. FLAG immunoprecipitations with anti-FLAG agarose affinity gel (Sigma M2) were performed followed by elution with 3xFLAG peptide (200 μg/ml from Sigma in the following buffer: 20 mM HEPES pH 7.5, 150 mM KCl, 0.1 mM EDTA, 10% glycerol, 0.1% Tween20, 1 mM DTT and supplemented with proteases, deacetylases, and phosphatase inhibitors), followed by Strep immunoprecipitation with Strep-Tactin Sepharose beads (IBA) and elution with 5 mM D-biotin in the same buffer used for FLAG elution.

The MORF and MOZ complexes subunits were visualized by SDS-PAGE followed by silver staining and confirmed by Western blotting using specific antibodies against MEAF6 (Ab42472), ING5 (ab96851) and Brd1 (Ab71877, recognizes both BRPF1 and BRPF2, see[29]). Fractions were also analyzed by mass spectrometry on an Orbitrap Fusion (Thermo Fisher) at the Proteomics Platform of the CHU de Québec Research Center, to identify bone fide subunits (Supplementary Data 1).

For purification of MORF complexes expressed from 293T cells (ATCC CRL-11268), plasmids of WT (ref. [24]) or F289A FLAG-MORF$_N$ together with HA-BRPF1, HA-MEAF6 and HA-ING5 were used to co-transfect 293T cells by the calcium phosphate method. Cells were harvested 48 h post-co-transfection and lysed in high-salt buffer (450 mM NaCl, 50 mM Tris-HCl (pH 8.0), 1% TX-100, 2 mM MgCl$_2$, 0.1 mM ZnCl$_2$, 2 mM EDTA, 10% glycerol) supplemented with protease inhibitor mixture. The NaCl concentration was reduced to 225 mM and the whole cell extract centrifuged at 14,000 rpm for 30 min. The FLAG MORF was purified from the soluble fraction using anti-FLAG M2 affinity beads (Sigma) for 2 h at 4 °C on rotating wheel. FLAG beads were next washed with the 225 mM NaCl buffer and eluted with 200 μg/ml of 3× FLAG peptide (Sigma).

In Fig. 4e and Supplementary Fig. 7, FLAG-tagged MORF proteins were overexpressed and prepared from transiently transfected 293T cells. Briefly, approximately 2 mg of total cell extract was incubated with 25 μl of M2-agarose beads (Sigma) overnight at 4 °C. The beads were washed four times with high-salt wash buffer (10 mM HEPES, pH 7.9, 25% glycerol, 1.5 mM MgCl$_2$, 300 mM KCl, and 0.1% Triton X-100), followed by two washes with low-salt wash buffer (10 mM HEPES, pH 7.9, 25% glycerol, 1.5 mM MgCl$_2$, 100 mM KCl, and 0.1% Triton X-100). Elution was achieved by two consecutive incubations of the beads with 0.5 mg/ml triple-FLAG peptide (Sigma) in 200 μl of low-salt wash buffer. All buffers contained phenylmethylsulfonyl fluoride and protease inhibitor cocktail (Roche).

The amounts of native (K562) or overexpressed (293T) MORF complexes were normalized empirically following SDS-PAGE and transfer onto nitrocellulose membrane. Anti-FLAG M2 conjugated to horseradish peroxidase (HRP; Sigma) was

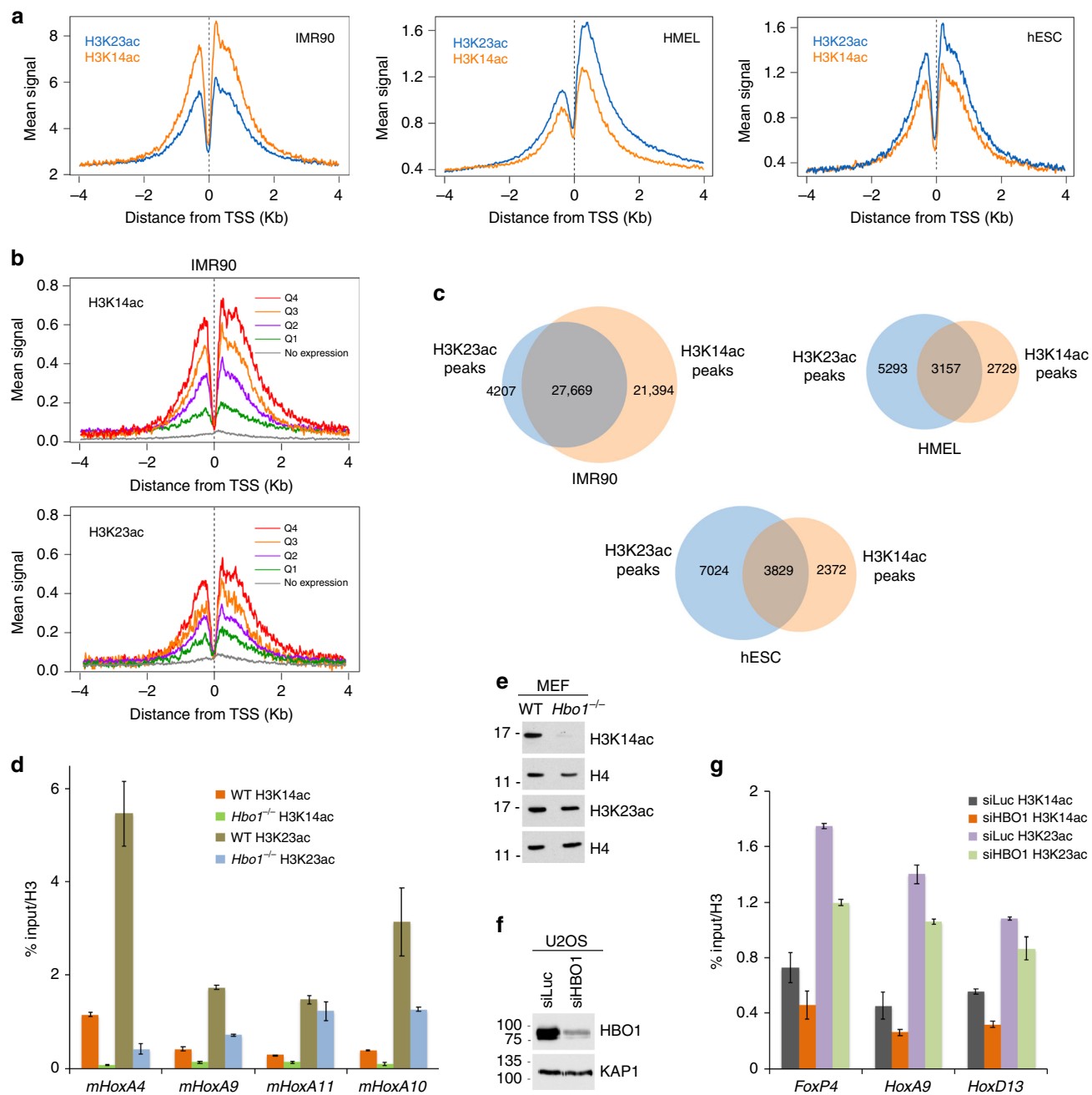

**Fig. 7** H3K23ac and H3K14ac colocalization and effect of H3K14ac on H3K23 acetylation at specific genes. **a** Metagene analysis of H3K14ac and H3K23ac signals around the transcription start sites of genes observed in ChIP-seq datasets derived from ENCODE in IMR90, hESC and HMEL cells. **b** RNA-seq analysis of H3K14ac and H3K23ac in IMR90 cell line derived from ENCODE. Genes were subdivided in quartiles corresponding to their expression levels. **c** Venn diagrams show overlap of the H3K14ac and H3K23ac peaks observed in IMR90, hESC and HMEL cells. **d** ChIP analysis of H3K23ac and H3K14ac in WT or *Hbo1*−/− MEFs. Acetylation levels were corrected for nucleosomes occupancy (total H3 signal). **e** Western blot analysis of histones purified from WT and *Hbo1*−/− MEFs with indicated antibodies. **f** Western blot analysis of HBO1 and KAP1 in U2OS cells treated with siLuciferase or siHBO1. **g** ChIP analysis of H3K23ac and H3K14ac in U2OS cells treated with siLuciferase or siHBO1. Acetylation levels were corrected for nucleosomes occupancy (total H3 signal). All ChIP values are based on two independent experiments and error bars in **d** and **g** indicate the range from duplicate samples, *n* = 2 independent experiments. Source data are provided as a Source Data file

used at a 1:10,000 dilution and the immunoblots were visualized using a Western Lightning plus-ECL reagent (Perkin-Elmer). Anti-HA-HRP (clone 3F10, Roche) was used at a 1:2000 dilution. Source data are provided as a Source Data file.

**HAT activity assays with purified MORF complexes.** Acetyltransferase activity of the purified complexes was measured with 0.125 μCi of ³H labeled Ac-CoA (2.1 Ci/mmol; PerkinElmer Life Sciences). The amount of the different purified MORF complexes used in the HAT assays were normalized based on the immunoblotting data (see Supplementary Fig. 1). The HAT reactions were performed in a volume of 15 μl using 0.5 μg of human free histones from HeLa cells or recombinant nucleosomes (NCP; Epicypher) or 0.3 μg of recombinant H3 peptides (residues 1–29, synthesized by us) as substrates, in HAT buffer (50 mM Tris-HCl (pH 8), 50 mM KCl, 10 mM sodium butyrate, 5% glycerol, 0.1 mM EDTA, 1 mM dithiothreitol) for 30 min at 30 °C. The reactions were then captured on P81 filter paper, the free H3-labeled Ac-CoA was washed away and the paper was analyzed using Liquid Scintillation.

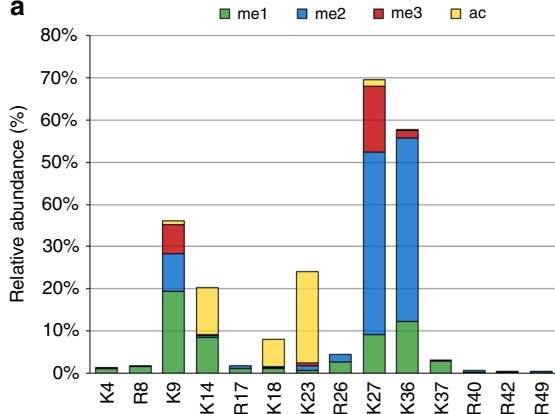

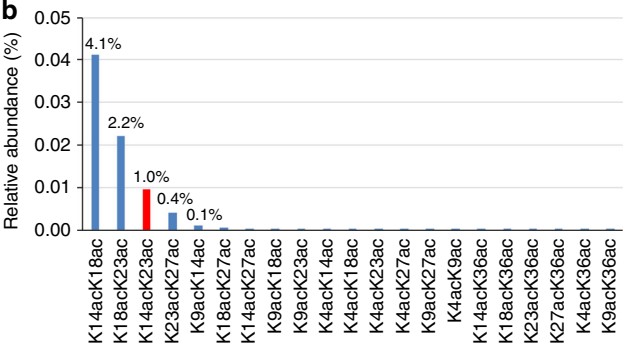

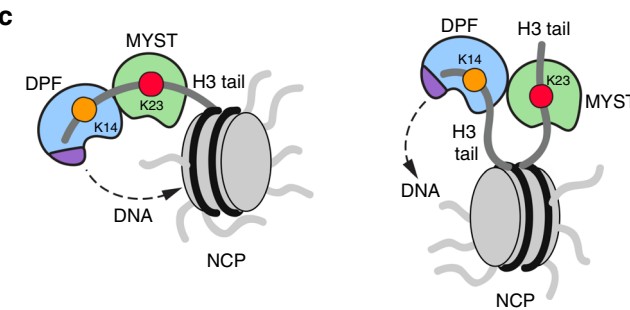

**Fig. 8** H3K23ac and H3K14ac co-exist. **a** Relative abundance of single histone PTMs quantified on the histone H3.2/1 N-terminal tail (residues 1–50). Color coding represents different PTM type, and the y-axis represents the relative abundance; the percentage missing to 100% is the abundance of the unmodified state. **b** Relative abundance of binary histone PTMs on histone H3. The percentage represents the total relative abundance of histone tails containing the two acetylations listed in the x-axis. The relative abundance of H3K14acK23ac is highlighted in red. **c** Models for the association of MORF_DPF-MYST *in cis* with a single H3 tail or *in trans* with two H3 tails of H3K14acyl-NCP

Alternatively, reactions were stopped by adding Laemmli sample buffer, run on SDS-PAGE, stained with coomassie, treated with EnH³ance, dried and expose on film for fluorography (Supplementary Fig. 1c).

In Fig. 4e and Supplementary Fig. 7, purified MORF proteins were incubated at 37 °C for indicated times with peptides (1 µg) and 0.1 mM acetyl-CoA in HAT reaction buffer (50 mM Tris, pH 8.0, 0.1 mM EDTA, 10% glycerol, 1 mM PMSF, and 1 mM DTT) in a total volume of 50 µL. Reactions were quenched by flash-freezing in liquid nitrogen and then analyzed by SDS-PAGE and Western blot.

**Protein purification**. MORF_DPF mutants (F287A, D289A, F218A, R276E, R306E/K309E, and F287A/R306E/K309E) were generated using the Quick Change site-directed lightning mutagenesis protocol (Stratagene) (Supplementary Table 2). Unlabeled and ¹⁵N-labeled WT and mutated MORF_DPF (residues 211–322) were expressed in *E. coli* Rosetta2 (DE3) pLysS cells grown in LB or NH₄Cl (Sigma-

Aldrich) minimal media supplemented with 50 µM ZnCl₂. After induction with IPTG (final concentration 0.5 mM) (Gold Biotechnology) for 16 h at 18 °C, cells were harvested via centrifugation and lysed by freeze-thaw followed by sonication. The unlabeled and ¹⁵N-labeled GST-fusion proteins were purified on glutathione Sepharose 4B beads (Thermo Fisher Science). The GST tag was either cleaved with Thrombin protease (MP Biomedicals), or left on and the proteins were eluted off the resin with 50 mM reduced L-glutathione (Fisher). The proteins were concentrated into PBS buffer pH 6.5, supplemented with 5 mM dithiothreitol (DTT). The unlabeled proteins were purified by size exclusion chromatography and concentrated in Millipore concentrators (Millipore).

**Peptide pulldown assay**. Histone peptides with different modifications were synthesized by CPC Scientific. For peptide pulldown assays, 1 µg of biotinylated histone peptides with different modifications were incubated with 1 µg of GST-fused proteins in binding buffer (50 mM Tris-HCl pH 7.5, 300 mM NaCl, 0.1% NP-40, 1 mM PMSF) overnight. Streptavidin beads (Amersham) were added to the mixture, and the mixture was incubated for 1 h with rotation. The beads were then washed three times and analyzed using SDS-PAGE and Western blotting.

**NMR spectroscopy**. ¹H,¹⁵N HSQC spectra were collected at 298 K on Varian INOVA 600 MHz and 900 MHz spectrometers. The binding was monitored by titrating H3 peptides (residues 1–19) into NMR samples containing 0.1–0.25 mM uniformly ¹⁵N-labeled WT or mutated MORF_DPF in 1×PBS pH 6.5 buffer supplemented with 5 mM DTT and 8–10% D₂O. NMR data were processed and analyzed as previously described[44]. The $K_d$ values were determined by a nonlinear least-squares analysis in Kaleidagraph using the following equation:

$$\Delta\delta = \Delta\delta_{max}\left(\left([L]+[P]+K_d\right)-\sqrt{\left([L]+[P]+K_d\right)^2-4[P][L]}\right)/2[P], \quad (1)$$

where $[L]$ is concentration of the peptide, $[P]$ is concentration of the protein, $\Delta\delta$ is the observed chemical shift change, and $\Delta\delta_{max}$ is the normalized chemical shift change at saturation. Normalized chemical shift changes were calculated using the equation

$$\Delta\delta = \sqrt{(\Delta\delta H)^2+(\Delta\delta N/5)^2}, \quad (2)$$

where $\Delta\delta$ is the change in chemical shift in parts per million (ppm).

**Fluorescence spectroscopy**. Spectra were recorded at 25 °C on a Fluoromax-3 spectrofluorometer (HORIBA). The samples containing 0.8–2 µM MORF_DPF in PBS buffer pH 6.5 supplemented with 5 mM DTT and progressively increasing concentrations of H3 peptides (residues 1–19) were excited at 295 nm. Emission spectra were recorded over a range of wavelengths between 310 and 405 nm with a 0.5 nm step size and a 0.6 s integration time. $K_d$ values were determined using a nonlinear least-squares analysis and the equation:

$$\Delta I = \Delta I_{max}\frac{\left(\left([L]+[P]+K_d\right)-\sqrt{\left([L]+[P]+K_d\right)^2-4[P][L]}\right)}{2[P]}, \quad (3)$$

where $[L]$ is concentration of the histone peptide, $[P]$ is concentration of the protein, $\Delta I$ is the observed change of signal intensity, and $\Delta I_{max}$ is the difference in signal intensity of the free and bound states of the protein. $K_d$ values were averaged over 2−4 separate experiments, and error was calculated as the standard deviation between the runs.

**X-ray crystallography**. Purified MORF_DPF (residues 211–322) was incubated at 10 mg/ml with the H3K14cr (residues 1–19) peptide (1:1.5 molar ratio) for 1 h on ice prior crystallization in 50 mM Tris-HCl pH 7.5 buffer, supplemented with 150 mM NaCl and 1 mM dithiothreitol. Crystals of the MORF_DPF-H3K14cr complex were grown using hanging-drop diffusion method at 18 °C by mixing protein-peptide solution with well solution composed of 1.1–1.3 M sodium citrate tribasic and 0.1 M sodium hepes pH 8.4–8.6 at a 1:1 ratio. The crystals were soaked in a mixture of 50% (1.1–1.3 M sodium citrate tribasic and 0.1 M sodium hepes pH 8.4–8.6) and 50−60% ethylene glycol prior to flash-freezing in liquid nitrogen. X-ray diffraction data were collected from a single crystal on the UC Denver X-ray Crystallography core facility Rigaku Micromax 007 high-frequency microfocus X-ray generator equipped with a Pilatus 200 K 2D area detector. Indexing and scaling were completed using HKL3000. The phase solution was found via molecular replacement using MORF_DPF-H3K14bu structure (PDB ID 5U2J) as a model. Model building was carried out with Coot[45] and refinement was performed with Phenix[46]. The final structure was validated by MOLProbity[47].

**MD simulation of the MORF_DPF-H3 complexes**. The crystal structures of the MORF_DPF-H3K14cr and MORF_DPF-H3K14bu complexes were used for modeling. AMBER ff14SB force field was used to describe the interaction energy potentials[48]. The atomic charges of the crotonyllysine and butyryllysine were determined by the RESP method[49]. Four zinc ions in the DPF were modeled with Zinc Amber Force Field (ZAFF)[50]. Each complex was immersed into the rhombic dodecahedron of

0.15 M KCl solution. The distance between the protein to the boundary of the solvent was at least 15 Å. The total number of atoms in the simulation was approximately 38,000. The MORF$_{DPF}$-H3 complex was generated by replacing crotonyllysine with unmodified lysine.

For each starting structure, 16 replicates of simulations were conducted. For each system, after the warming-up run of 1.1 ns, 250 ns long simulation was conducted under the NPT (constant temperature and pressure) condition. The trajectories of initial 100 ns of the simulation were discarded as the equilibrium run from analyses. The time step for the simulation was set to 2 fs with constraining bonds involving hydrogens. Simulations were performed with GROMACS 2016.

**Electrophoretic mobility shift assay.** A total of 32 repeats of the 601 Widom DNA (147 bp) sequence were cloned into the pJ201 plasmid and transformed into DH5α cells. The plasmid was purified, the individual sequences were separated by digestion with EcoRV, and the 601 Widom DNA was purified from the remaining plasmid by gel extraction (Qiagen-MinElute Gel Extraction kit). Increasing amounts of WT or mutated GST-MORF$_{DPF}$ were incubated with 601 Widom DNA or H3K14cr-NCP (0.5 pmol/lane) in buffer (20 mM Tris-HCl pH 7.5,100 mM NaCl, 0.2 mM EDTA, and 20% glycerol) for up to 10 min on ice. The reaction mixtures were loaded on 5% native polyacrylamide gels and electrophoresis was performed in 0.2× TBE buffer (1× TBE = 90 mM Tris, 64.6 mM boric acid, and 2 mM EDTA) at 80–100 V on ice. Gels were stained with SYBR Gold (Thermo Fisher) and visualized by Blue LED (UltraThin LED Illuminator- GelCompany).

**Expression and purification of histone H3K14ac and H3K14cr.** To incorporate N-ε-acetyl-L-lysine (AcK) into the K14 position of histone H3, pETDuet-1 vector encoding human histone H3 with amber stop codon (pETDuet-1-H3K14TAG) introduced at H3K14 was used to cotransform E. coli BL21 (DE3)-ΔcobB strain with pEVOL-MmAcKRS. Single colony was picked and inoculated in 2YT medium supplemented with 100 mg/l ampicillin and 34 mg/l chloramphenicol. When OD reached to 0.6, H3 expression was induced at 37 °C by adding 0.5 mM IPTG, 0.2% (w/v) L-arabinose and 5 mM AcK into cell culture. Cells were harvested 6 h after induction, and purified in the same steps as previously reported[51]. The whole procedure of CrK (N-ε-crotonyl-L-lysine) incorporation into the K14 position of histone H3 was identical with AcK incorporation, except that pEVOL-MmPylRS-384W vector was used, and 1 mM of CrK (N-ε-crotonyl-L-lysine) was added into 2YT medium after induction.

**Assembly of nucleosomes.** Recombinant Histone His-TEV-H2A, His-TEV-H2B and His-SUMO-TEV-H4 were purified in the previously reported steps[52]. All four histone pellets including histone H3K14ac or H3K14cr were dissolved in 6 M GuHCl buffer (6 M guanidinium hydrochloride, 20 mM Tris, 500 mM NaCl, pH 7.5), and concentration was measured by UV absorption at 280 nm (Biotek synergy H1 plate reader). To prepare H2A/H2B dimer, His-TEV-H2A and His-TEV-H2B were mixed in the molar ratio of 1:1, and 6 M GuHCl buffer was added to adjust total protein concentration to 4 μg/μl. Denatured His-TEV-H2A/His-TEV-H2B solution was dialyzed sequentially at 4 °C against 2 M TE buffer (2 M NaCl, 20 mM Tris, 1 mM EDTA, pH 7.8), 1 M TE buffer (1 M NaCl, 20 mM Tris, 1 mM EDTA), 0.5 M TE buffer (0.5 M NaCl, 20 mM Tris, 1 mM EDTA). Then the resulting dimer solution was centrifuged for 5 min at 4 °C to remove precipitates, and the concentration of dimer was determined by UV absorption at 280 nm. Steps of His-TEV-H3/His-SUMO-TEV-H4 tetramer refolding were generally the same as His-TEV-H2A/His-TEV-H2B dimers except that the total protein concentration should be adjusted to 2 μg/μl and no stirring in the sequential dialysis. Then His-TEV-H2A/Hi-TEV-H2B dimers were mixed with His-TEV-H3/His-SUMO-TEV-H4 tetramers in a molar ratio of 1:1 to generate histone octamers, and NaCl solid was added to adjust NaCl concentration to 2 M. The 147 bp biotinylated 601 nucleosome positioning was prepared by polymerase chain reaction with biotinylated primers and purified by PCR cleanup kit (#2360250 Epoch Life Science). Purified 147 bp DNA was re-dissolved in 2 M TE buffer and added to histone octamer solution in the molar ratio of 0.85:1. 2 M TE buffer was added to adjust final 147 bp DNA concentration to 2–3 μM. The DNA histone mixture solution was then transferred to a dialysis bag and placed inside about 200 ml 2 M TE buffer, while stirring at room temperature, Tris buffer with no salt (20 mM Tris) was slowly added into the 2 M salt buffer through a liquid transfer pump (#23609-170 VWR®). Nucleosomes formed when salt concentration was reduced to around 150 mM (measured by EX170 salinity meter), and the DNA histone mixture solution was further dialyzed into low-salt Tris buffer (20 mM Tris, 20 mM NaCl, 0.5 mM EDTA, pH 7.8). Precipitates were removed by centrifuge, and the concentration of the nucleosomes was measured by A260 reading using the Biotek synergy H1 plate reader. His-TEV protease was added to nucleosome solutions (TEV:nucleosome 1:30, w-w) to remove all the histone tags after incubation for 1 h at 37 °C, and finally all the His tagged impurities were removed from nucleosome solution by Ni$^{2+}$-NTA resin (Thermo Fisher #88221).

**ChIP assays.** Chromatin preparation from MEF, U2OS (ATCC HTB-96) and 293T cells were done as previously described[53]. For immunoprecipitation of chromatin, we used 300 μg of chromatin (for FLAG ChIP, FLAG-M2 from Sigma)

and 15–50 μg of chromatin (for histone ChIP, H3K14ac (07–353) and H3K23ac (07–355) from Millipore, H3 (ab1781) from Abcam) with 1–3 μg of specific antibodies incubated overnight at 4 °C. Next, 50 μl of Protein G Dynabeads for FLAG ChIP or 25 μl of Protein A Dynabeads was added to each sample and incubated for 4 h at 4 °C. The beads were washed extensively and eluted with 1% SDS and 0.1 M NaHCO3. Cross-linked samples were reversed by heating overnight at 65 °C in the presence of 0.2 M NaCl. Samples were then treated with RNase A and proteinase K for 2 h, and DNA was recovered by phenol-chloroform and ethanol precipitation. Quantitative real-time PCR corrected for primer efficiencies in the linear range were performed using SYBR Green I (Roche) on a LightCycler 480 (Roche).

**ChIP-seq analysis.** Raw ChIP-seq data of H3K14ac, H3K23ac in HMEL cell line and BRPF1/2 and ING5 in RKO cell line were downloaded from the GEO database (GSE47190 and GSE58953). The ChIP-seq accession numbers used in the UCSC profiles were GSM1146448, GSM1146449, GSM1146452, GSM1422905, GSM1422906 and GSM1422908. Processed RNA-seq data of H3K14ac, H3K23ac in IMR90 cell line were downloaded from the ENCODE project (accession number: ENCFF694APR). Raw ChIP-seq data of H3K14ac, H3K23ac in hESC and IMR90 cell line were downloaded from the ENCODE project (accession numbers for H3K14ac: ENCSR283STF and ENCSR057BTG; accession numbers for H3K23ac: ENCSR942NME and ENCSR815LBP). The raw reads were mapped to human reference genome NCBI 36 (hg19) by Bowtie 1.1, allowing up to two mismatches. The ChIP-seq profiles were generated using model-based analysis of ChIP-seq (MACS) (v.2.1) with parameters --nomodel and --bdg. The profile signals were subtracted by H3 or Input control data. Clonal reads were automatically removed by MACS. The ChIP-seq profiles were drawn using the UCSC Custom tracks utility. For the Venn diagrams, peaks are overlapping if any peaks are overlapping or near within 1000 bp. The ChIP-seq profiles for H3K14ac and H3K23ac are centered around TSSs of RefSeq genes with ±4000 bp extensions. The ChIP-seq data used in the plots were normalized to 30,000,000 total tag numbers.

**HBO1 knockdown and knockout cells.** HBO1 knockdown was performed in U2OS cells by using siRNA HBO1 (Dharmacon)[10] and siLuciferase (Doyya) was used as a control. Transfection was performed with lipofectamine RNAiMAX transfection reagent (Life Technologies) following the manufacturer's instructions, and cells were collected after 66 h. Hbo1$^{+/+}$ and Hbo1$^{-/-}$ mouse embryonic fibroblasts were described in ref.[37], immortalized using a shRNA against Trp53 and fully controlled.

**MS analysis of histone H3 PTMs.** Raw data were obtained from Schwammle et al.[40], in which histones were extracted from mouse embryonic stem cells wild type, Suz12$^{-/-}$, Ring1A/B$^{-/-}$ and Dnmt triple knockout (strain E14). Histones were digested with GluC at a ratio of 1:50 enzyme:sample in 50 mM ammonium bicarbonate overnight at room temperature. Obtained intact histone tails (aa 1–50) were analyzed as previously described[54]. Briefly, histones were loaded and separated using a two column system consisting on a 5 cm pre-column (100 μm ID) packed with C$_{18}$ bulk material (ReproSil, Pur C18AQ 5 μm; Dr. Maisch) and a 22 cm analytical column with pulled needle (75 μm ID) packed with Polycat A resin (PolyLC, Columbia, MD, 3 μm particles, 1500 Å). Loading buffer was 0.1% formic acid. Buffer A was 75% acetonitrile, 20 mM propionic acid (Fluka), adjusted to pH 6.0 using ammonium hydroxide (Sigma-Aldrich), and solvent B was 25% acetonitrile adjusted to pH 2.5 with formic acid. Histones were run at least in four replicates at a flowrate of 250 nl/min, with a gradient of 5 min 100% solvent A, followed by 55−85% solvent B in 150 min. Data were acquired in an LTQ-Orbitrap Velos with ETD source (Thermo Scientific) was coupled online with the nanoLC used for separation (Dionex Ultimate 3000). The acquisition was performed in the Orbitrap for both precursors and products, with a resolution of 60,000 (full-width at half-height) for MS and 30,000 for MS/MS. Precursor charge state 1+, 2+ and 3 + were excluded. Isolation width was set at 2 m/z. The six most intense ions were isolated for fragmentation using ETD with an activation Q value of 0.25, activation time of 20 ms, and supplementary activation. m/z window was set at 450–750 to include only charge states 8–10. The acquisition method was set for not using dynamic exclusion.

Spectra were then deconvoluted with Xtract (Thermo Scientific) and searched using Mascot (v2.5, Matrix Science, London, UK) with the following search parameters: MS mass tolerance 2.2 Da; MS/MS tolerance 0.01 Da; enzyme GluC with 0 missed cleavages allowed, database containing only histone H3 sequences (updated October 2012). Variable modifications were: mono- and dimethylation (KR), trimethylation (K) and acetylation (K). Raw files and annotated spectra are available at the ProteomeXchange database (Accession: PXD002560). Identified spectra were filtered and quantified using in-house software isoScale slim[55], available at http://middle-down.github.io/Software.

**Reporting summary.** Further information on research design is available in the Nature Research Reporting Summary linked to this Article.

## Data availability

The atomic coordinates and structure factors of MORF$_{DPF}$-H3K14cr have been deposited in the Protein Data Bank under the accession code 6OIE. Raw mass spec data for MORF and MOZ have been deposited to the MassIVE database, ID number MSV000084304. All other relevant data supporting the key findings of this study are available within the Article and its Supplementary Information files or from the corresponding authors upon reasonable request. The source data underlying Figs. 1d, 2b, 2c, 2e, 7e and 7f are provided as a Source Data file. A Reporting Summary for this Article is available as  a Supplementary Information file.

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

## Acknowledgements
We thank X.J. Yang for kindly providing the cDNA of wild-type FLAG-MORF$_N$ and R.A. Dickins for kindly providing the *Trp53* shRNA knockdown construct. This work was supported by NIH grants R01 GM100907, GM106416 and GM125195 to T.G.K., CA204020 to X.S., GM126900 to B.D.S., R01GM121584 to W.R.L., and GM110174 and CA196539 to B.A.G., by JSPS KAKENHI JP18H05534 and Platform Project for Supporting Drug Discovery and Life Science Research (BINDS) from AMED JP18am0101106j0002, JP19am0101106j0003 to H.K., by Canadian Institutes of Health Research (CIHR FDN-143314) to J.C., Welch Grant A-1715 to W.R.L., and Leukemia & Lymphoma Society Career Development Award (1339-17) to X.S. and by the Australian Government's National Health and Medical Research Council through Project Grants 1084352 (to A.K.V. and T.T.) and 1084248 (A.K.V. and T.T.) and Research Fellowships (to A.K.V. 575512 and 1081421; and to T.T. 1003435), and by the Independent Research Institutes Infrastructure Support (IRIIS) Scheme and a Victorian State Government OIS (Operational Infrastructure Support) Grant. J.C. holds the Canada Research Chair in Chromatin Biology and Molecular Epigenetics.

## Author contributions
B.J.K., S.M.J., C.L., W.M., J. Lyu, S. Sakuraba, K.K., W.W.W., S. Sidoli, J. Liu, Y.Z., X.W., B. M.W. and A.J.K. performed experiments and together with A.K.V, T.T., B.A.G., W.R.L., B.D.S., H.K., W.L., X.S., J.C. and T.G.K. analyzed the data. B.J.K., J.C. and T.G.K. wrote the manuscript with input from all authors.

## Competing interests
The authors declare no competing interests.
