## [Peer Review File · Nature Communications]

Reviewers' comments:

Reviewer #1 (Remarks to the Author):

In this manuscript, Klein et al identify the MORF complex, containing a catalytic MYST family acetyltransferase domain, as a histone H3K23 acetyltransferase. This has been achieved by affinity purification of the complex, followed by its testing on H3 peptides that carry acetylation at specific positions. Activity assay show that unmodified H3 is a better substrate than K14 acetylated or crotonylated H3 peptide (Figure 1e, native complex), and that there is a comparable activity for H3 and H3K14cr, with somewhat reduced activity on H3K14ac peptide (Figure 1f, reconstituted complex). Comparison between K14ac and K14cr in both data sets suggests potential dual role of K14: that while K14 modification in both cases likely blocks MORF acetylation at this particular site, K14cr can also stimulate the activity of the complex relative to acetylation. To investigate if the DPF motif of MORF, the domain that was previously shown to recognize H3K14cr, may responsible for the stimulation, authors have deleted this domain and determined that MORFdeltaDPF indeed has reduced activity relative to the WT MORF, both on histones and to the lesser degree on short oligonucleosomes (Fig 1g). Together, these findings support the conclusion that the double PHD finger (DPF) domain of the catalytic MORF subunit positively regulates catalytic activity, and acetylation by the complex is coupled to binding of DPF to its H3K14 crotonylated peptide ligand.

Authors have further assessed affinity of H3WT, K14ac and K14cr peptides for the DPF domain, demonstrating a fivefold increase in binding affinity in the presence of crotonylation, and attenuated 2x increase with acetylation. The structural basis of recognition was established by a co-crystal structure of DPF with H3K14cr peptide, showing conservation of acyllysine binding mechanisms compared to that observed with the same domain and K14bu peptide, as well as DPF domain of KAT MOZ, in complex with H3K14cr. Further mutagenesis has implicated a Phe residue in stabilization of crotonyl-Lys binding. Additionally, presence of a positively charged surface has prompted the authors to investigate if DPF domain can bind DNA, showing that it does and that K209 and R306 residues are involved in binding, both in the context of free DNA and in the context of a reconstituted H3K14cr-modified nucleosome core particle. Further mutational analysis has contributed to parsing out the roles of the individual residues in binding to histone and to DNA.

To investigate the importance of DPF domain of MORF in H3K23 acetylation, the authors tested acetylation ability of the reconstituted complex harboring mutations that either abrogate crotonylation recognition selectivity (F287A) or enhance its recognition (triple FRK mutant). With peptides as substrates, F287A MORF complex has attenuated activity both with H3 and H3K14cr peptides, consistent with the decreased binding affinity of H3 peptides. While FRK mutant has somewhat decreased activity relative to the WT, this mutant preserves crotonylation preference. Presumably heterogeneous, with respect to H3K14 modification, core histone and short oligonucleosomes (SONs) substrates generally hold these trends, albeit the effects on SONs are attenuated. Interpretation of these findings however is challenging without knowledge on K14 modification. Experiments with reconstituted nucleosomes and MORF complexes support role of acylation in stimulating acetylation activity.

Analysis of genome wide-localization of K14ac and K23ac marks (Fig 6) shows a substantial overlap between the two modifications. Co-occupancy of MORF subunit at these sites is consistent with the model where MORF catalyzes H3K14ac-coupled H3K27ac to activate transcription. Further support for the model is provided by mass spec analysis of co-frequency of H3K14 and K27 acetylation (Figure 7).

Altogether, the study provides strong support for MORF complex as H3K23 acetylation catalyst, and supports the model where H3K14 acylation promotes H3K23 modification by the complex by engaging DPF domain. Co-frequency of the marks, combined with co-occupancy of MORF subunits at H3K14acK23ac marked sites lends further support to the model. These findings are significant

as they not only identify MORF as the regulator of K23 acetylation, but also uncover how its activity is regulated by chromatin context. However, the manuscript does require further strengthening to unambiguously support these claims. These points are highlighted below:

Major:

- Given decreased activity on K14ac peptide and residual activity with K23ac peptide, it will be important to determine if indeed K14 can be acetylated by the MORF complex through mass spec analysis of MORF treated H3 peptide and mapping of modifications.
- To further evaluate the model, it is critical to uncouple two function of K14 residue – potential acetylation site as well as the site of stimulatory crotonylation. Use of H3K14R mutant peptide, in addition to K14 and K23 modified peptides, would eliminate K14 as a possible modification site in the H3 peptide. It would be important to test how MORFdeltaDPF construct behaves on the panel of H3 peptides used in Figure 1e and 1f, in addition to H3K14R.
- H3K14cr binding mechanism. An SI figure showing: a) superposition of two structures (K14cr and K14bu), (b) the channel discussed and (c) the binding pocket of the crotonyl/butyryl group would help follow the discussion. Particularly important is to compare positioning of F218 in both to evaluate pi-pi stacking. Binding data for K14bu is missing, and would further strengthen this evaluation. Also, what is the B factor for I267 and S217? Finally, the reasoning for S242 mutation needs further clarification. It should be labeled in the figure. To support the claim that S217 and S242 are important for crotonyl selectivity, binding data for K14bu and K14ac should be provided.
- Errors should be provided for all Trp fluorescence data.
- Is weak DNA binding relevant (Fig 4)? Also, in EMSA assays, only DNA disappears. Assays needs further improvements or it should be removed as it is not clear whether it is relevant and what it adds.
- Line 231: The reasoning behind the triple mutant should be explained and the corresponding figure cited.
- Without further characterization of substrates, it is unclear how to interpret data in Figure 5b. Similarly, how do authors rationalize enhanced acetylation of H3K14ac NCP in the assay with crotonylation non-discriminatory mutant F287A (Fig 5c)? Is the effect of this mutation on recognition of acetylation known?
- Lines 132-133: "The HAT activity of the MORFdeltaDPF complex was substantially compromised...., indicating that DPF affects the catalytic function" The compromised activity may also result from the deletion. Additional support, using a mutant that abrogates DPF binding, is needed.
- It is interesting that nucleosomes are substrates for FRK mutant MORF complex, despite inability of R306E/K309E mutant to bind DNA. Does this data suggest that in the context of the MORF complex DNA recognition is not dependent on DPF domain? Are there other components of the complex that recognize DNA? Overall, DNA binding does not seem relevant to the activity of the complex.
- Inability of FRK mutant DPF calls into question DNA interactions proposed in the conclusion and in figure 7c.
- Hypothesized cis vs trans arrangement should be tested by supplementing HAT activity assay on H3 peptide (Figure 5a, WT, black bar) with H3K14crK23R peptide.

Minor:

- It would be useful to add MORFN construct into Figure 1.
- Fig 1e: as control signal is very low, controls are difficult to associate with grey bars. Could Y axis be modified to zoom in on low signal that region?
- Line 144: "3 fold tighter... (Figure 2b.c). The robust association" Can this be confidently claimed, with error of 0.8 for H3K14ac peptide and Kd of 1.9?
- Please provide information on exact nature of delta DPF deletion (which residues have been removed)?
- p. 11, line 239: is the difference between H3 and H3K14cr statistically significant. If not, "even more reduced" should be removed.
- Ramachandran outliers missing in Table S2

Reviewer #2 (Remarks to the Author):

Reviewer Comments

In this manuscript, Klein and colleagues identify that the MORF complex is an acyltransferase for H3K23. They uncover the biochemical regulation of the MORF complex through dissecting its ability to bind both nucleosomes and DNA. The authors begin by purifying native MORF complexes for downstream use in HAT assays on a series of modified peptides. They demonstrate that HAT activity is most robust on unmodified H3 peptides and is lowest on H3K23 sites—suggesting H3K23 is likely being modified by MORF. They further demonstrate that the DPF of MORF is partially required for enzymatic activity of MORF. The authors then probe DPF binding different modified H3 peptides. They discover that the DPF of MORF has the strongest binding affinity for H3K14cr. They then solve the structure of MORF in complex with H3K14cr. They characterize this binding event and generate mutants that abolish binding. The authors demonstrate that F218 forms pi-pi interactions with the crotonyl moiety, and that this interaction can explain the increased binding affinity over acetyl-lysine. The authors demonstrate MORF can bind DNA, and that this DNA binding enhanced the overall binding affinity in modified nucleosome core particles. They attempt to demonstrate, using mutant MORF complexes, that these binding events are critical for enzyme activity. Finally, the authors demonstrate that MORF co-occupies chromatin marked by H3K23ac and H3K14ac. They also demonstrate that these two histone modifications co-exist with relatively high stoichiometry in vivo. The described story is interesting. Some comments are described below.

Major:

- Figure 1E, what is “control” represent? What is in this reaction? Is it just Ac-CoA without enzyme? Please clarify.
- Figure 2 is interesting finding presented here in, in which the authors demonstrate acyl preference for H3K14cr. However, a comparison of the binding affinity to structurally similar acyl-lysine modification, lysine butyrylation, should have been performed in this assay.
- Figure 3 attempts to describe the preference in binding Kcr modification. While the authors in the manuscript do mention Kbu, they proceed to compare the binding of Kcr to Kac, despite the major structural differences in these modifications relative to the similarities shared between Kbu and Kcr. This same group demonstrated that Kbu binds with 1.2 uM affinity while Kac binds with 1.9 uM affinity, citing hydrophobic interactions that mediate this difference in binding (Klein BJ, Simithy J, Wang X, et al. Recognition of Histone H3K14 Acylation by MORF. *Structure*. 2017;25(4):650–654.e2. doi:10.1016/j.str.2017.02.003). Figure3 A to C: this experiment should include Kbu peptide.
- Why did the authors choose to establish whether MORF has DNA binding activity? While these experiments are not trivial, and do reveal something about the biochemical nature of MORF, the rationale for this finding is not clear. Nor is it in the title, or the last line of the abstract where the authors present their model.
- Figure 1F and figure 5A: WT MORF HAT activity in figure 5a is not consistent with WT HAT activity in Figure 1F. There’s a great deal of variability in this system. I am not convinced that binding of Kcr is important for enzyme activity.
- Figure 5B: the effect of MORF mutants on enzyme activity toward native short oligonucleosomes is minor. some explanation should be provided.

Minor:

- For figure 1, the authors claim H3K23ac is blocking the catalytic activity of MORF (line 113-115). This makes it seem as if MORF is being sterically impaired or inhibited through regulation. Enzymatic/catalytic activity may still well be active (or perhaps not, however, this was not tested).

The system merely lacks substrate.

- Suggest to include a few other acyl-lysine modifications and use them to compare the Kds of binding.

Reviewer #3 (Remarks to the Author):

In this study, the authors reported a structural mechanism explaining histone acetyltransferase MORF's selective HAT activity for histone H3 lysine 23 acetylation (H3K27ac). Specifically, they demonstrated that the H3K23ac specificity by the HAT domain of MORF is positively regulated by the DPF domain of MORF that is adjacent to the HAT domain and prefers to bind acylated lysine 14 in histone H3. They also presented the crystal structure of the DPF domain in complex with a crotonylated H3K14 peptide, which provides the molecular details of this selective molecular recognition. They further provided genomic sequencing evidence for the co-existence of H3K23ac and H3K14ac at promoters of MORF target genes in cells. The authors concluded that MORF works through this crosstalk mechanism involving two acylation modifications in histone H3 to regulate gene transcriptional activation in chromatin. Overall, this is an interesting study. However, some of the conclusions do not seem to be well supported by the results presented in the manuscript as follows.

1. From the in vitro biochemical and structural analyses, the MORF DPF domain appears to prefer binding to H3K14cr over H3K14ac. However, there was almost no cellular evidence provided in this study that would verify this MORF DPF domain/H3K14cr interaction in a biologically relevant setting.
2. The MORF DPF domain's ability to bind to histone H3 (by comparing WT with F287A or FRK mutants) showed limited added value to overall HAT activity of MORF using either histone H3 peptide, SON or even H3K14ac-bearing NCP. This would argue against the role of DPF domain in positively contributing to the HAT activity or specificity of MORF. It will be important for the authors to include histone H3 peptides or NCP that contain other known H3 lysine acetylation sites, such as H3K9ac, H3K18ac and H3K27ac. This would demonstrate whether or how H3K14cr/ac is selective in positive contribution to H3K23ac by MORF HAT domain.
3. To further prove the role of MORF DPF domain binding to H3K14cr/ac, the authors should consider profiling WT and mutants of the DPF domain on MORF HAT activity on a few target genes in cells.
4. The genomic sequencing data presented in Figure 6 do not really support the proposed mechanism of the role of H3K14cr in positive regulation of H3K23ac by MORF. At minimum, the role of H3K14ac/cr needs to be experimentally validated by ChIP-qPCR for a number of MORF target genes such as those shown in Figure 6d.
5. From the data presented in Figure 4, it was not clear whether MORF DPF domain binding to DNA is selective, or simply non-specific electrostatic interactions. Does DPD domain have any preference for single-strand or double-strand DNA, or even RNA? The loss of the signal in EMSA for DNA and protein mixtures could also be due to possible loading problem, which needs to be ruled out. Also, the authors need to obtain quantitative measurements of DPF domain and DNA binding by the fluorescence anisotropy method. Furthermore, the question about possible functional importance of DPF/DNA binding was not addressed in this study. The FRK mutant data presented in Figure 5 indicated that the DNA binding was not important for MORF HAT activity on H3K23ac.
6. Minor,
no figure legend was provided for Figure 4j. In the legend of Figure 4, (f) should be (k).

Reviewer #4 (Remarks to the Author):

In this manuscript the authors characterize the enzymatic activity of the MORF acetyltransferase. Specifically, they show that MORF displays a preference for lysine 23 of the H3 histone tail and that acylation of a nearby lysine (K14) slightly enhances acetylation of K23. The study focuses on characterizing (structure and function) a region of MORF called DPF that binds the H3 tail. They solve the crystal structure of MORFDPF in complex with a crotonylated H3K14 peptide and compare this with previously solved structures with other acylated H3 peptides. Existing ENCODE data was used to show that in cells the H3K23ac and H3K14ac modifications co-exist with MORF. With these correlations and in vitro biochemical data suggesting MORF can bind DNA, the authors conclude that interaction of MORFDPF with acylated H3K14 promotes acetylation of H3K23 by the native MORF complex to activate gene transcription. While this study presents a number of interesting observations on MORF acetyltransferase activity, collectively the study falls short of providing convincing evidence for their model. Several experiments are not rigorously quantified, small differences are often interpreted as highly important for the model, while moderate differences in values are interpreted as more dramatic than the number would predict. The DNA binding experiments complicate the model, and in fact appear to contradict the idea that K14 acylation matters, leading the reader a bit confused on how all of these observations fit neatly into the model presented. In conclusion, the manuscript is of potential interest to the chromatin field, but in its current state is an incomplete story with some technical concerns.

1.) The results section is filled with non-quantitative terms to describe results. Throughout the manuscript terms such as "still robust yet less than," "robust" "robustly," "was nearly abolished," "barely," "almost no," and "blocked" are written instead of actual numbers with statistical consideration. A careful look at the actual data presented in the figures often suggest a different interpretation. Rigorous quantification needs to be included so that the reader can more accurately assess the results. In figure 1e, the HAT activity of FL MORF is highest for the unmodified H3 peptide, and data with the pre-acetylated K23 peptide suggests that MORF can still acetylate K14 with only a 4-5 fold lower efficiency. The controls clearly show extremely low background, so the results with K14 are very significant, and not "nearly abolished" as written in the results section. Figure 1f shows similar results. A more accurate conclusion is that both K14 and K23 are substrates for MORF with a ~5 fold preference for K23. Figure 1g suggests that the DPF contributes very little if at all to acetylation of histones and nucleosomes. The relative activity on histones appears higher than on nucleosomes. The controls are not defined in Figure 1g, and there seems to be two different types of controls, again not defined.

2.) The data in figure 2 suggests the DPF binds crotonyl groups with ~2-3 fold tighter affinity than the unmodified H3 peptide, but this is not uniformly true. There appears to be reduced binding when the peptide is acetylated in the pull down assay while the Trp fluorescence data doesn't fully agree. There appears to be greater discrimination when K27cr vs K27ac are compared. The authors claim that F218 is the major driving force for crotonyl binding, but the data are not compelling, especially when the errors in measurements are considered. At best, this is about 2-3 fold. In fact one of the values was reported from a prior publication, suggesting that the experiments were done in a prior experiment, not controlling for 'batch' effects.

3.) In figure 3, the authors make a structure and dynamic simulation argument for the role of F218. Have the authors calculated the free energies with and without F218, and do these agree with the extremely small 2-3 fold effect on binding observed in the biochemical experiments. One would predict there to be a major change in binding affinity if F218 were to function as suggested by the structure, but the actual biochemical data do not support this. Perhaps a different model is needed.

4.) The data in figure 4 marks a major departure from the prior results. Here the authors investigate DNA binding. The EMSAs show loss of free DNA but there are no other new bands formed, so interpreting such experiments without additional corroborating evidence is problematic.

This whole analysis does not connect well with Figures 1-3. There are no experiments in Fig 4 that show the binding depends on crotonyl groups. No controls with acetyl or unmodified are included. The data showing the loss of DNA binding with the basic residue mutations is very interesting, but may have nothing to do with K14acylation.

5.) A clear interpretation of the activity data presented in Figure 5 would be challenging. The effects are subtle and similar to those in Figure 1 (comment 1), but with additional mutation data.

6.) The analysis of the ENCODE data showing the co-occupancy is a nice addition, though the data are not unexpected, regardless of the proposed mechanism whereby K14acylation enhances K23 acetylation by MORF. Similarly, the middle down proteomics shows the 1% co-existence of K14 and K23 acetylation, which is a nice addition, but does not provide a mechanistic explanation. Also, the idea that crotonylation is preferred is not supported by this analysis.

7.) This study has potential, but will need a major revision to address the technical concerns and the authors should consider a possible re-focus on the DNA binding or possibly the stronger discrimination at K27 (fig 2a), but regardless the paper needs a cohesive and focused set of results that leads to a very clear model. The connection between crotonyl selectivity and DNA binding is confusion. This reviewer is not sure the subtle acyl selectivity at K23 is the most compelling of the results presented, but if the authors want to continue with this model, then more causal experiments should be performed in cells and in vitro. Perhaps the authors could demonstrate in cells that MORF collaborates with p300/CBP or GCN5 only after these enzymes crotonylate or acetylate K14.

We thank the Editor and Reviewers for the insightful and very constructive comments, which were helpful in revising and strengthening this manuscript.

In the revised manuscript, new data are shown in Figs. 1e, 1g; 2c (F218A/H3K14bu); 4a, 4b (right panels), 4c; 6a, 6b, 6c; 7d, 7e, 7f, 7g; Suppl. Figs. 1d; 3; 4; 6a; 10.

Reviewer 1, Comment 1: These findings are significant as they not only identify MORF as the regulator of K23 acetylation, but also uncover how its activity is regulated by chromatin context. Given decreased activity on K14ac peptide and residual activity with K23ac peptide, it will be important to determine if indeed K14 can be acetylated by the MORF complex through mass spec analysis of MORF treated H3 peptide and mapping of modifications.

Author's response: to clarify that it was previously reported that MORF can acetylate H3K14, we have added the following sentence and citations (page 4): "Specifically, MOZ/MORF complexes were reported to acetylate H3K14 and H4K5/8/12/15 *in vitro* and H3K9 *in vivo*⁷⁻¹²."

Reviewer 1, Comment 2: ... To further evaluate the model, it is critical to uncouple two function of K14 residue – potential acetylation site as well as the site of stimulatory crotonylation. Use of H3K14R mutant peptide, in addition to K14 and K23 modified peptides, would eliminate K14 as a possible modification site in the H3 peptide. It would be important to test how MORFdeltaDPF construct behaves on the panel of H3 peptides used in Figure 1e and 1f, in addition to H3K14R.

Author's response: as suggested, we have tested MORFdeltaDPF (new data are shown in Fig. 1e). The deletion of DPF results in a decrease in HAT activity of the MORF complex. For the K14R mutation, because this mutation not only eliminates acetylation on K14 but also decreases binding of DPF, we will not be able to properly compare HAT activities of MORF.

Reviewer 1, Comment 3: ... H3K14cr binding mechanism. An SI figure showing: a) superposition of two structures (K14cr and K14bu), (b) the channel discussed and (c) the binding pocket of the crotonyl/butyryl group would help follow the discussion. Particularly important is to compare positioning of F218 in both to evaluate pi-pi stacking. Binding data for K14bu is missing, and would further strengthen this evaluation. Also, what is the B factor for I267 and S217? Finally, the reasoning for S242 mutation needs further clarification. It should be labeled in the figure. To support the claim that S217 and S242 are important for crotonyl selectivity, binding data for K14bu and K14ac should be provided.

Author's response: as suggested, the superimposed structures of MORF_{DPF} in complex with H3K14cr and H3K14bu are now shown in Suppl. Fig. 4. We have also included new tryptophan fluorescence binding data for H3K14bu in Fig. 2c and Suppl. Fig. 3. B factor for S217 and I267 is 20/24 and 30/32, chains A/B respectively. We do not show S217D/S242R data in the revised manuscript because the idea and rationale for mutating the corresponding residues in homologous MOZ is nicely and thoroughly described in the Xiong et al, NCB, 2016 study.

Reviewer 1, Comment 4: Errors should be provided for all Trp fluorescence data. – all Trp fluorescence data have errors. All binding curves are now shown in Suppl. Fig. 3.

Reviewer 1, Comment 5: Is weak DNA binding relevant (Fig 4)? Also, in EMSA assays, only DNA disappears.

Author's response: our data indicate that it is relevant: binding of the DPF domain to NCP was substantially decreased when DNA-binding residues were mutated (Fig. 5d). The shift of DNA

band is seen in Fig. 5a. Of note, we found that DNA complexes are often less visible in EMSA for readers that have weak DNA-binding activity (YEATS, PZP, PHD, Tudor).

Reviewer 1, Comment 6: ... Line 231: The reasoning behind the triple mutant should be explained and the corresponding figure cited.

Author's response: The following sentence has been added to clarify the reasoning (page 11): "Conversely, replacing two positive charges with negative charges near the H3 binding pocket (mutating R306 and K309 to Glu) resulted in a ~3-fold increase in binding to H3K14cr peptide by the F287A/R306E/K309E mutant compared to the F287A mutant (Fig. 4a and b, left panels, and Suppl. Fig. 6b)."

Reviewer 1, Comment 7: Without further characterization of substrates, it is unclear how to interpret data in Figure 5b. Similarly, how do authors rationalize enhanced acetylation of H3K14ac NCP in the assay with crotonylation non-discriminatory mutant F287A (Fig 5c)? Is the effect of this mutation on recognition of acetylation known?

Author's response: Figure 5b has been removed. Although histone binding function of F287A mutant is decreased, it still has residual binding activity and associates better with acetylated (including acetylated) H3K14 peptide than with unmodified H3 peptide. As suggested, we have carried out NMR titration experiments with the F287A mutant and H3K14ac and H3 peptides and calculated K_d s to show the effect of acetylation (new Fig. 4a, b, right panels and Suppl. Fig. 6a).

Reviewer 1, Comment 8: ... Lines 132-133: "The HAT activity of the MORF Δ DPF complex was substantially compromised...., indicating that DPF affects the catalytic function" The compromised activity may also result from the deletion. Additional support, using a mutant that abrogates DPF binding, is needed.

Author's response: we agree, this sentence has been revised to: "The HAT activity of the MORF Δ DPF complex was decreased ~2-fold on the H3 and H3K23ac peptides and on purified free histones, and ~4-fold on the H3K14ac peptide, indicating that DPF affects the catalytic function of the MORF complex." We tested the loss of histone binding activity F287A mutant in HAT assays to confirm that the histone binding function of DPF is necessary (Fig. 4c-e). We also carried out CHIP assays with this mutant (Fig. 6a-c).

Reviewer 1, Comment 9: It is interesting that nucleosomes are substrates for FRK mutant MORF complex, despite inability of R306E/K309E mutant to bind DNA. Does this data suggest that in the context of the MORF complex DNA recognition is not dependent on DPF domain? Are there other components of the complex that recognize DNA? ...

Author's response: the HAT assays with the FRK mutant have been removed to avoid confusion and the following sentence is added on page 15: "It will also be important to investigate the relationship between the MORF and BRPF1 subunits of the MORF complex, as the PZP domain of BRPF1 also binds to H3 tail and DNA³⁴."

Reviewer 1, Comment 10: Hypothesized cis vs trans arrangement should be tested by supplementing HAT activity assay on H3 peptide (Figure 5a, WT, black bar) with H3K14crK23R peptide.

Author's response: because both K14 and K23 are unavailable for acetylation in the H3K14crK23R peptide, there should be almost no HAT activity of the complex on this peptide.

Reviewer 1, Comment 11: It would be useful to add MORFN construct into Figure 1.- **added**
- Fig 1e: as control signal is very low, controls are difficult to associate with grey bars. Could Y axis be modified to zoom in on low signal that region? - **this figure has been removed**
- Line 144: “3 fold tighter... (Figure 2b.c). The robust association” Can this be confidently claimed, with error of 0.8 for H3K14ac peptide and Kd of 1.9? – **agree, this phrase has been removed**
- Please provide information on exact nature of delta DPF deletion (which residues have been removed)?- **included, page 15**
- p. 11, line 239: is the difference between H3 and H3K14cr statistically significant. If not, “even more reduced” should be removed. – **agree, removed**
- Ramachandran outliers missing in Table S2 – **added, thank you**

Reviewer 2, Comment 1: ... Figure 1E, what is “control” represent? What is in this reaction? Is it just Ac-CoA without enzyme? Please clarify..

Author’s response: this figure has been removed.

Reviewer 2, Comment 2: Figure 2 is interesting finding presented here in, in which the authors demonstrate acyl preference for H3K14cr. However, a comparison of the binding affinity to structurally similar acyl-lysine modification, lysine butyrylation, should have been performed in this assay...

Author’s response: as suggested, new tryptophan fluorescence binding data for H3K14bu are now included in Fig. 2c and Suppl. Fig. 3.

Reviewer 2, Comment 3: Why did the authors choose to establish whether MORF has DNA binding activity? While these experiments are not trivial, and do reveal something about the biochemical nature of MORF, the rationale for this finding is not clear...

Author’s response: we have clarified the rationale on page 10.

Reviewer 2, Comment 4: ... Figure 1F and figure 5A: WT MORF HAT activity in figure 5a is not consistent with WT HAT activity in Figure 1F. There’s a great deal of variability in this system. I am not convinced that binding of Kcr is important for enzyme activity.

Author’s response: we have revised Figure 5a and also added cartoon in Fig 1g to highlight that unmodified K14 is a substrate for both domains, MYST and DPF (however acylation of K14 enhances binding of DPF).

Reviewer 2, Comment 5: Figure 5B: the effect of MORF mutants on enzyme activity toward native short oligonucleosomes is minor. some explanation should be provided.

Author’s response: these data have been removed.

Reviewer 2, Comment 6: For figure 1, the authors claim H3K23ac is blocking the catalytic activity of MORF (line 113-115). This makes it seem as if MORF is being sterically impaired or inhibited through regulation. Enzymatic/catalytic activity may still well be active (or perhaps not, however, this was not tested). The system merely lacks substrate.

• Suggest to include a few other acyl-lysine modifications and use them to compare the Kds of binding.

Author's response: we agree, this sentence has been revised to: "The MORF complex acetylated unmodified H3 peptide, but its catalytic activity was considerably, by ~12-fold, decreased on the H3 peptide that was pre-acetylated at Lys23 (H3K23ac) (Fig. 1e, blue bars)." We have compared H3K14ac, H3K14su and H3K14hib in Klein, Structure 2017.

Reviewer 3, Comment 1: From the in vitro biochemical and structural analyses, the MORF DPF domain appears to prefer binding to H3K14cr over H3K14ac. However, there was almost no cellular evidence provided in this study that would verify this MORF DPF domain/H3K14cr interaction in a biologically relevant setting.

Author's response: it is indeed very challenging to delineate the effects of crotonylation and acetylation in cells. We cite the studies by Allis and Li groups (Xiong, NCB 2016, Sabari, MolCell 2015, refs. 30, 32), which describe meticulously performed analysis that reveals differences in biological outcomes associated with these two acyl marks.

Reviewer 3, Comment 2: The MORF DPF domain's ability to bind to histone H3 (by comparing WT with F287A or FRK mutants) showed limited added value to overall HAT activity of MORF using either histone H3 peptide, SON or even H3K14ac-bearing NCP. This would argue against the role of DPF domain in positively contributing to the HAT activity or specificity of MORF. It will be important for the authors to include histone H3 peptides or NCP that contain other known H3 lysine acetylation sites, such as H3K9ac, H3K18ac and H3K27ac. This would demonstrate whether or how H3K14cr/ac is selective in positive contribution to H3K23ac by MORF HAT domain.

Author's response: we have revised Fig. 4, removed the FRK HAT data, and added the following sentence on page 10: "A less pronounced reduction in the HAT activity of the MORF_N F287A complex on recombinant NCP and H3K14ac-NCP (Fig. 4d, green and orange bars) indicated that other readers present in the complex, including the PZP domain of BRPF1 that binds to H3 tail and DNA³⁴, tether the complex to NCPs, counterbalancing the loss of the interaction with MORF_{DPF}." We and others have previously shown that DPFs interact with the first 15 residues of H3 and acylation of only K14 but not K9, K18 or K27 enhances this interaction (Zeng, Nature 2010, Dreveny, NAR 2013, Qiu, G&D 2012, Ali, JMB 2012, Xiong, NCB 2016).

Reviewer 3, Comment 3: To further prove the role of MORF DPF domain binding to H3K14cr/ac, the authors should consider profiling WT and mutants of the DPF domain on MORF HAT activity on a few target genes in cells. ... The genomic sequencing data presented in Figure 6 do not really support the proposed mechanism of the role of H3K14cr in positive regulation of H3K23ac by MORF. At minimum, the role of H3K14ac/cr needs to be experimentally validated by ChIP-qPCR for a number of MORF target genes such as those shown in Figure 6d.

Author's response: as suggested, we have performed ChIP assays with WT and mutant MORF to show the importance of histone binding by DPF for binding of MORF to promoters of target genes and their acetylation on H3K23 (Fig. 6a-c). The new data (Fig. 7d-g) also demonstrate that loss of H3K14ac at specific genes leads to loss of H3K23ac.

Reviewer 3, Comment 4: From the data presented in Figure 4, it was not clear whether MORF DPF domain binding to DNA is selective, or simply non-specific electrostatic interactions. Does DPD domain have any preference for single-strand or double-strand DNA, or even RNA? The loss of the signal in EMSA for DNA and protein mixtures could also be due to possible loading problem, which needs to be ruled out. Also, the authors need to obtain quantitative measurements of DPF domain and DNA binding by the fluorescence anisotropy method. Furthermore, the question about possible functional importance

of DPF/DNA binding was not addressed in this study. The FRK mutant data presented in Figure 5 indicated that the DNA binding was not important for MORF HAT activity on H3K23ac.

Author's response: a number of histone readers (YEATS, PZP, some PHDs, Tudors, BDs) have DNA binding activity, it's a non-specific electrostatic contact that increases binding of readers to chromatin. Despite being relatively weak, the interactions with DNA are essential. Here, we show that binding of the DPF domain to NCP was substantially decreased when DNA-binding residues were mutated (Fig. 5d). All full EMSA gels are shown in Suppl. Fig. 10. The HAT assay data with the FRK mutant have been removed.

Reviewer 3, Comment 5: no figure legend was provided for Figure 4j. In the legend of Figure 4, (f) should be (k).

Author's response: figure legend is now included (Suppl. Fig. 6c).

Reviewer 4, Comment 1: 1.) The results section is filled with non-quantitative terms to describe results. Throughout the manuscript terms such as “still robust yet less than,” “robust” “robustly,” “was nearly abolished,” “barely,” “almost no,” and “blocked” are written instead of actual numbers with statistical consideration. A careful look at the actual data presented in the figures often suggest a different interpretation. Rigorous quantification needs to be included so that the reader can more accurately assess the results. In figure 1e, the HAT activity of FL MORF is highest for the unmodified H3 peptide, and data with the pre-acetylated K23 peptide suggests that MORF can still acetylate K14 with only a 4-5 fold lower efficiency. The controls clearly show extremely low background, so the results with K14 are very significant, and not “nearly abolished” as written in the results section. Figure 1f shows similar results. A more accurate conclusion is that both K14 and K23 are substrates for MORF with a ~5 fold preference for K23. Figure 1g suggests that the DPF contributes very little if at all to acetylation of histones and nucleosomes. The relative activity on histones appears higher than on nucleosomes. The controls are not defined in Figure 1g, and there seems to be two different types of controls, again not defined.

Author's response: we agree and revised the text to include quantifications: “The MORF complex acetylated unmodified H3 peptide, but its catalytic activity was considerably, by ~12-fold, decreased on the H3 peptide that was pre-acetylated at Lys23 (H3K23ac) (Fig. 1e, blue bars). In contrast, HAT activity of the complex on the peptide pre-acetylated at Lys14 (H3K14ac) was reduced only ~1.2-fold. ... The HAT activity of this complex was only slightly, by ~1.2-fold, decreased on H3K14ac peptide, implying that H3K14 is not its primary substrate. Again, a substantial ~7-fold decrease in the catalytic activity of the complex was observed on H3K23ac peptide, supporting the idea that H3K23 is the major target for acetylation (Fig. 1g)..... The HAT activity of the MORF ΔDPF complex was decreased ~2-fold on the H3 and H3K23ac peptides and on purified free histones, and ~4-fold on the H3K14ac peptide, indicating that DPF affects the catalytic function of the MORF complex.”

Fig. 1g (now Suppl. Fig. 2) has been revised and controls are defined.

Reviewer 4, Comment 2: 2.) The data in figure 2 suggests the DPF binds crotonyl groups with ~2-3 fold tighter affinity than the unmodified H3 peptide, but this is not uniformly true. There appears to be reduced binding when the peptide is acetylated in the pull down assay while the Trp fluorescence data doesn't fully agree. There appears to be greater discrimination when K27cr vs K27ac are compared. The authors claim that F218 is the major driving force for crotonyl binding, but the data are not compelling, especially when the errors in measurements are considered. At best, this is about 2-3 fold.

In fact one of the values was reported from a prior publication, suggesting that the experiments were done in a prior experiment, not controlling for 'batch' effects.

Author's response: It was previously shown that DPFs bind to the first 15 residues of H3 (Ala1-Arg2 of H3 are strictly required) and not to the residues 22-44 of H3 (Zeng, Nature 2010, Dreveny, NAR 2013, Qiu, G&D 2012, Ali, JMB 2012, Xiong, NCB 2016, Klein, Structure, 2017). These studies also report a similar ~2-4 fold enhancement in binding due to acetylation of H3K14. We have removed the second pulldown gel to avoid confusion.

Reviewer 4, Comment 3: 3.) In figure 3, the authors make a structure and dynamic simulation argument for the role of F218. Have the authors calculated the free energies with and without F218, and do these agree with the extremely small 2-3 fold effect on binding observed in the biochemical experiments. One would predict there to be a major change in binding affinity if F218 were to function as suggested by the structure, but the actual biochemical data do not support this. Perhaps a different model is needed.

Author's response: We have performed MD simulations of the MORF_{DPF}-H3K14bu complex and found that butyryllysine shuttles between two conformations (unmodified H3 is incapable of doing so). These data suggest that the ping-pong like acyllysine binding mechanism might be a characteristic feature of the MORF_{DPF}-H3K14acyl complex formation, but not MORF_{DPF}-H3 complex formation. We have revised the text on page 9 accordingly.

We haven't calculated the free energies, but we measured K_d for the interaction of F218A mutant with H3K14bu. The F218A mutation eliminates selectivity of DPF for H3K14cr as the F218A mutant binds equally well to acetylated, butyrylated and crotonylated H3K14 (Fig. 2c).

Reviewer 4, Comment 4: 4.) The data in figure 4 marks a major departure from the prior results. Here the authors investigate DNA binding. The EMSAs show loss of free DNA but there are no other new bands formed, so interpreting such experiments without additional corroborating evidence is problematic. This whole analysis does not connect well with Figures 1-3. There are no experiments in Fig 4 that show the binding depends on crotonyl groups. No controls with acetyl or unmodified are included. The data showing the loss of DNA binding with the basic residue mutations is very interesting, but may have nothing to do with K14acetylation.

Author's response: the DNA binding part has been revised, shortened, and moved to Fig. 5. The shift of DNA band is seen in Fig. 5a. Of note, we found that DNA complexes are often less visible in EMSA for readers that have weak DNA-binding activity (YEATS, PZP, PHD, Tudor). Fig. 5c shows a decrease in binding of the DPF mutant with impaired histone binding activity to H3K14cr-NCP.

Reviewer 4, Comment 5: 5.) A clear interpretation of the activity data presented in Figure 5 would be challenging. The effects are subtle and similar to those in Figure 1 (comment 1), but with additional mutation data.

Author's response: we have revised the text (pages 9, 10), quantified the data, and added cartoon (Fig. 1g) to better understand the results. The following sentences have been added: "The catalytic activity of the full length MORF F287A complex on free histones was reduced two-fold compared to the HAT activity of the native WT MORF complex, indicating that functional MORF_{DPF} is essential (Fig. 4c and Suppl. Fig. 1d). ... A less pronounced reduction in the HAT activity of the MORF_N F287A complex on recombinant NCP and H3K14ac-NCP (Fig. 4d, green and orange bars) indicated that other readers present in the complex, including the

PZP domain of BRPF1 that binds to H3 tail and DNA³⁴, tether the complex to NCPs, counterbalancing the loss of the interaction with MORF_{DPF}.”

Reviewer 4, Comment 6: 6.) The analysis of the ENCODE data showing the co-occupancy is a nice addition, though the data are not unexpected, regardless of the proposed mechanism whereby K14acylation enhances K23 acetylation by MORF. Similarly, the middle down proteomics shows the 1% co-existence of K14 and K23 acetylation, which is a nice addition, but does not provide a mechanistic explanation. Also, the idea that crotonylation is preferred is not supported by this analysis.

Author's response: we have added new *in vivo* data (Figs. 6a-c and 7d-g) that demonstrate the role of DPF in MORF binding to genes and stimulation of local H3K23ac. We also show that depletion of HBO1-dependent H3K14ac leads to the decrease of H3K23ac at specific genes.

Reviewer 4, Comment 7: 7.) This study has potential, but will need a major revision to address the technical concerns and the authors should consider a possible re-focus on the DNA binding or possibly the stronger discrimination at K27 (fig 2a), but regardless the paper needs a cohesive and focused set of results that leads to a very clear model. The connection between crotonyl selectivity and DNA binding is confusion. This reviewer is not sure the subtle acyl selectivity at K23 is the most compelling of the results presented, but if the authors want to continue with this model, then more causal experiments should be performed in cells and *in vitro*. Perhaps the authors could demonstrate in cells that MORF collaborates with p300/CBP or GCN5 only after these enzymes crotonylate or acetylate K14.

Author's response: as suggested, we have performed *in vivo* experiments, including ChIP-qPCR, KO and KD HBO1 (to demonstrate MORF/HBO1 collaboration) (shown in Fig. 6a-c and Fig. 7d-g), added *in vitro* HAT data (Fig. 1e and Fig. 4c), revised the DNA binding part and the HAT activity part and clarified the rationale for studying the DNA binding (many readers bind both histone tails and DNA, therefore the DPF domain is a new addition to the set of readers with dual binding capabilities; but DPF has a unique setting of the closely positioned histone- and DNA-binding sites that hasn't been observed in any other currently known reader).

REVIEWERS' COMMENTS:

Reviewer #1 (Remarks to the Author):

The authors have addressed all of the comments from the initial review. The revised manuscript is well put together, clear, and of substantial interest to the community.

Reviewer #2 (Remarks to the Author):

The authors have addressed all the questions i raised and i suggest the paper as it is.

Reviewer #3 (Remarks to the Author):

The authors have successfully addressed my previous comments. The current manuscript is recommended for publication.

Reviewer #4 (Remarks to the Author):

The authors have greatly improved the manuscript based on the original concerns. This is a very interesting study that would appeal to researchers in the chromatin field studying the mechanisms of reading and modifying histones and nucleosomes. I appreciate the authors including more quantitative descriptors so the reader can reach their own conclusions, even if these are different than the authors'.

John Denu

We thank the Editor and Reviewers for the insightful and very constructive comments, which were helpful in revising and strengthening this manuscript.

Reviewer #1 (Remarks to the Author):

The authors have addressed all of the comments from the initial review. The revised manuscript is well put together, clear, and of substantial interest to the community.

Reviewer #2 (Remarks to the Author):

The authors have addressed all the questions i raised and i suggest the paper as it is.

Reviewer #3 (Remarks to the Author):

The authors have successfully addressed my previous comments. The current manuscript is recommended for publication.

Reviewer #4 (Remarks to the Author):

The authors have greatly improved the manuscript based on the original concerns. This is a very interesting study that would appeal to researchers in the chromatin field studying the mechanisms of reading and modifying histones and nucleosomes. I appreciate the authors including more quantitative descriptors so the reader can reach their own conclusions, even if these are different than the authors'.
John Denu